# MONTY HALL AND OPTIMIZED CONFORMAL PREDICTION TO IMPROVE DECISION-MAKING WITH LLMS

## ABSTRACT

Large language models (LLMs) are empowering decision-making in open-world agents in several applications, including tool or API usage and answering multiple choice questions (MCQs). However, they often make overconfident, incorrect predictions, which can be risky in high-stakes settings like healthcare and finance. To mitigate these risks, recent works have used conformal prediction (CP), a model-agnostic framework for distribution-free uncertainty quantification. CP transforms a *score function* into prediction sets that contain the true answer with high probability. While CP provides this coverage guarantee for arbitrary scores, the score quality significantly impacts prediction set sizes. Prior works have relied on LLM logits or other heuristic scores, lacking quality guarantees. We address this limitation by introducing CP-OPT, an optimization framework to learn scores that minimize set sizes while maintaining coverage. Furthermore, inspired by the Monty Hall problem, we extend CP's utility beyond uncertainty quantification to improve accuracy. We propose a method called *conformal revision of questions* (CROQ) to revise the problem by narrowing down the available choices to those in the prediction set. The coverage guarantee of CP ensures that the correct choice is in the revised question prompt with high probability, while the smaller number of choices increases the LLM's chances of answering it correctly. Experiments on the MMLU, ToolAlpaca, and TruthfulQA datasets with Llama-3 and Phi-3 models show that optimized CP scores reduce set sizes while maintaining coverage guarantee, and CROQ shows significant improvement in accuracy over the standard inference procedure.

## 1 INTRODUCTION

Large language models (LLMs) (Touvron et al., 2023; Databricks, 2024; Abdin et al., 2024) have demonstrated remarkable capabilities in various decision-making tasks, including multi-choice question answering (MCQ) and tool usage, where the model must select the correct tool or API to complete a task (Qu et al., 2024; Tang et al., 2023; Hendrycks et al., 2021). However, quantifying the uncertainty of an LLM's predictions remains challenging, with LLMs themselves often exhibiting overconfidence in wrong answers (Krause et al., 2023; Groot and Valdenegro Toro, 2024). Given a reliable uncertainty measure, the LLM could, in highly uncertain instances, take measures such as deferring to a human or invoking a secondary model, thereby increasing overall accuracy. Without this, it may be difficult to prevent the LLM from making serious errors in these settings. To address these challenges, we focus on improving uncertainty quantification (UQ) and its utility in MCQ and tool usage tasks.

Conformal prediction (CP) is one framework that can be used to quantify uncertainty in LLMs (Kumar et al., 2023; Su et al., 2024; Mohri and Hashimoto, 2024; Quach et al., 2024). Conformal prediction is a model-agnostic and distribution-free technique for producing prediction sets that contain the true outcome with a user-specified probability (e.g. 95%). The set sizes provide a natural measure of uncertainty, with small (large) sets representing low (high) uncertainty. The construction of sets in CP depends on a *score function*, which measures how well a candidate output "conforms" to a given input. For example, in classification settings, it is common to use the classifier logits corresponding to each class for a given input as the score function (Angelopoulos and Bates, 2022). While conformal prediction gives a coverage guarantee for *any* score function, the size of the output sets depends on the score function that is used. As an example, a random score function will yield

output sets that constitute random subsets of the label space that are large enough to satisfy the coverage guarantee (Angelopoulos and Bates, 2022).

Previous works that apply conformal prediction for MCQ type settings have used readily available scores such as the logits (softmax) output from the LLM (Kumar et al., 2023) or have designed heuristic scores based for example on repeated querying of the LLM (Su et al., 2024). Logits can be overconfident and may show biases for some options (Zheng et al., 2024), and heuristic scores are not guaranteed to produce small sets. Thus we need a principled solution to obtain scores that are designed to minimize set sizes (uncertainty) while preserving the coverage guarantee.

In addition to providing a measure of uncertainty, we believe that the sets produced by conformal prediction can be leveraged for other downstream purposes. Specifically, inspired by the Monty Hall problem (Selvin, 1975; Rosenthal, 2008), we hypothesize that by revising a multiple-choice question or tool usage prompt to include only the options within a conformal prediction set, the LLM is more likely to provide the correct answer due to the reduced number of choices, leading to improved accuracy.

To summarize, we focus on (1) obtaining optimal scores for conformal prediction for MCQ and tool usage tasks in LLMs, and (2) investigating whether the prediction sets generated by conformal prediction can be used to improve downstream accuracy. Our main contributions are as follows:

1. We design a score function optimization framework (CP-OPT) that can be applied to any pre-trained LLM. Moving away from the potentially unreliable LLM logits and heuristic scores, our framework provides a principled way to learn the scores for conformal prediction. Empirically, we show that our procedure leads to reduction in average set sizes in contrast to the baseline procedure that uses the LLM logits as the scores, at the same level (95%) of marginal coverage.

2. Extending the utility of conformal prediction beyond uncertainty quantification, we propose the conformal revision of questions (CROQs), in which we revise the question by narrowing down the choices to those in the prediction sets output by conformal prediction. Then the LLM is re-prompted with the revised question. We show that this procedure can boost overall accuracy by upto $5\%$ (Figure 2) and upto 14% on part of ToolAlpaca dataset with 10 options (Table 2).

## 2 PRELIMINARIES

In this section, we introduce notation and provide background on solving MCQ tasks with LLMs and conformal prediction.

### 2.1 SOLVING MULTIPLE CHOICE QUESTIONS (MCQS) USING LLMS

**MCQ Setup.** MCQs are a general abstraction for expressing problems in which the correct choice(s) must be selected from a given set of choices. These encompass conventional question-answering tasks such as MMLU (Hendrycks et al., 2021) as well as other tasks such as tool learning, in which the LLM must select the correct tool or API to complete a task (Tang et al., 2023; Qu et al., 2024). An MCQ consists of the question text $Q$, i.e. a sequence of tokens, and a set of answer choices $O = \{(Y_1, V_1), (Y_2, V_2), \ldots, (Y_m, V_m)\}$. Here, each $Y_j$ is a unique character from the English alphabet, and we assume that the number of choices $m$ is less than or equal to the size of the alphabet. Each $V_j$ is the option text for the $j$th option. Denote the whole MCQ instance as $x = (Q, O)$. Let $\mathcal{X}_m$ denote the space of MCQs with $m$ choices and $\mathbb{P}_{\mathcal{X}_m}$ denote a distribution over $\mathcal{X}_m$, from which samples for training, calibration, and testing are drawn independently. Here, we assume that for each question $Q$ there is only one correct answer key $y^\star \in \{Y_1, Y_2, \ldots Y_m\} = \mathcal{Y}_m$.

**MCQ Prompt.** We concatenate the question text $Q$ and the answer choices $O$, all separated by a new line character, and append to the end the text "The correct answer is: ". The expectation is that given this input prompt, the next token predicted by the LLM will be one of the option keys. See Appendix D for a prompt example. We consider zero-shot prompts and do not include example questions and answers in the prompt. We also add the prefix and suffix tokens to the prompt as recommended by the language model providers. Since these are fixed modifications to $x$, we will use $x$ to denote the final prompt and the MCQ instance analogously.

**LLM Inference.** We run the forward pass of the auto-regressive LLM (Touvron et al., 2023; Dubey et al., 2024; Abdin et al., 2024) on the input prompt to obtain the logit scores for each possible next token given the prompt, restricting attention to the tokens that correspond to the available answer keys (e.g. "a", "b", "c", "d" if there are four answer options). We take the softmax to convert the logits to probabilities, and then we take as the LLM's answer the option with the highest probability. This approach ensures that the LLM's answer will be one of the available answer options, which would not be guaranteed if instead we asked the LLM to simply generate an answer token given the prompt. This approach mirrors what has been done in other works that use LLMs to solve MCQs (Kumar et al., 2023; Su et al., 2024). Formal details are given in Appendix A.

## 2.2 CONFORMAL PREDICTION

Conformal prediction (CP) (Vovk et al., 2005; Angelopoulos et al., 2022) is a framework for quantifying uncertainty in machine learning models. It provides a flexible and user-friendly approach to output *prediction sets* (which may be finite sets or intervals) which contain the true output or label with a probability that is specified by the user, e.g. $95\%$. The key strength of conformal prediction lies in its *distribution-free* guarantees: it ensures that the constructed prediction sets are valid regardless of the underlying data distribution and model. This property is particularly desirable in the context of language models, as it is hard to characterize language data distributions or put specific distributional assumptions/restrictions on the LLMs.

**Score Function.** Let $g : \mathcal{X}_m \times \mathcal{Y}_m \mapsto \mathbb{R}$ be a conformal *score function*, where larger scores indicate better agreement ("conformity") between $x$ and $y$. Intuitively, large scores are intended to indicate that $y$ is a plausible output given $x$, while smaller scores indicate less plausibility. (Note that some authors prefer to have larger scores indicate greater disagreement, e.g. Clarkson et al. (2024).) A common choice of score function is the softmax scores from the given model. For closed-source LLMs, where these scores are not available, other authors have devised self-consistency scores based on repeated querying of the model (Su et al., 2024).

**Prediction Sets.** Given a threshold $\tau$ on the scores, the prediction set for any $x \in \mathcal{X}_m$ is given by

$$C(x; g, \tau) := \{y \in \mathcal{Y}_m : g(x, y) \geq \tau\}. \tag{1}$$

Intuitively, larger sets represent greater uncertainty, while smaller sets represent less uncertainty. This can be used for example to compare two different score functions given a fixed confidence level: a score function that produces larger sets can be said to result in greater uncertainty. Note that because the coverage guarantee is marginal over the data distribution (Proposition 2.1), it does not immediately follow that set sizes can be compared across different parts of the input space: coverage conditional on specific set sizes is not necessarily equal to the overall coverage level. However, previous work has found that the size of sets correlates with LLM accuracy in an MCQ setting (Kumar et al., 2023), indicating the set size as a measure of confidence in the LLM's output.

**Split Conformal Prediction.** Similar to prior works (Kumar et al., 2023; Su et al., 2024), we use *Split Conformal Prediction* (Papadopoulos et al., 2002; Lei et al., 2018) due to its popularity, ease of use, and computational efficiency. Given a score function $g : \mathcal{X}_m \times \mathcal{Y}_m \mapsto \mathbb{R}$, Split Conformal Prediction uses a calibration dataset $D_{\mathrm{cal}} = \{x_i, y_i^\star\}_{i=1}^{n_{\mathrm{cal}}}$ to compute a threshold $\hat{\tau}$, defined as

$$\hat{\tau} = \min \left\{ q : \frac{1}{n_{\mathrm{cal}}} \sum_{i=1}^{n_{\mathrm{cal}}} \mathbb{1}\left(g(x_i, y_i^\star) \leq q\right) \geq \alpha \right\}, \tag{2}$$

where $\alpha \in [0, 1]$ is a user-chosen error rate that is equal to 1 minus the desired coverage; for example, a value of $\alpha = 0.05$ would correspond to a coverage of $95\%$. In words, $\hat{\tau}$ is the smallest empirical quantile of the scores for the correct answers on the calibration dataset that is sufficient to satisfy (an empirical version of) the coverage property. The threshold $\hat{\tau}$ is used to construct prediction sets $C(x; g, \hat{\tau})$ on previously unseen test points as in (1). This procedure enjoys a marginal coverage guarantee for prediction sets constructed on unseen test data points, formalized as Proposition 2.1.

**Proposition 2.1.** *(Marginal Coverage Guarantee) (Lei et al., 2018, Thm. 2.2) Let $g$ be a fixed ~~conformity~~ score function and $\hat{\tau}$ be an $\alpha$ threshold computed via Split Conformal Prediction on $D_{\mathrm{cal}} = \{x_i, y_i^\star\}_{i=1}^{n_{\mathrm{cal}}} \sim \mathbb{P}_{\mathcal{X}_m \times \mathcal{Y}_m}$. Then, for a new sample $(\tilde{x}, \tilde{y}^\star) \sim \mathbb{P}_{\mathcal{X}_m \times \mathcal{Y}_m}$, we have that*

$$\mathbb{P}(\tilde{y}^\star \in C(\tilde{x}; g, \hat{\tau})) \geq 1 - \alpha. \tag{3}$$

*where the probability is marginal over the randomness in the calibration data and the new sample.*

The top half of Figure 1 illustrates conformal prediction for answering MCQs with LLMs. While the coverage guarantee in Proposition 2.1 holds for any score function, ideally we would like a score function that yields the smallest sets possible (the least uncertainty). Next, we discuss our solutions to improve conformal prediction and its utility in solving MCQs with LLMs.

## 3 METHODOLOGY

In this section, we discuss details of our method for learning optimal scores for conformal prediction and our pipeline for question revision using conformal prediction.

### 3.1 SCORES OPTIMIZATION FOR CONFORMAL PREDICTION (CP-OPT)

We describe our method for learning the optimal scores for conformal prediction (CP) for solving MCQs with LLMs. Similar ideas have been incorporated in the training objective of classifiers (Stutz et al., 2022) so that the classifiers' softmax output is better suited for CP. However, the LLMs are not trained with this objective, and we want to apply CP to any given LLM; therefore, we design a post-hoc method to optimize the scores. We first characterize the optimal scores and then describe how we can estimate them in practice.

#### 3.1.1 CHARACTERIZATION OF THE OPTIMAL SCORES

For any score function $g : \mathcal{X}_m \times \mathcal{Y}_m \mapsto \mathbb{R}$ and threshold $\tau$, the membership of any $y$ in the prediction set $C(x \mid g, \tau)$ is given by $\mathbb{1}(y \in C(x \mid g, \tau)) \iff \mathbb{1}\{g(x, y) \geq \tau\}$. Define the expected set size $S(g, \tau)$ and the coverage conditional on $\tau$, denoted $\mathcal{P}(g, \tau)$, as follows:

$$S(g, \tau) := \mathbb{E}_x \Big[ \sum_{y \in \mathcal{Y}_m} \mathbb{1}\{g(x, y) \geq \tau\} \Big] = \sum_{y \in \mathcal{Y}_m} \mathbb{E}_x \left[ \mathbb{1}\{g(x, y) \geq \tau\} \right], \quad (4)$$

$$\mathcal{P}(g, \tau) := \mathbb{E}_x \left[ \mathbb{1}\{g(x, y^\star) \geq \tau\} \right]. \quad (5)$$

The optimal score function $g^\star$ and threshold $\tau^\star$ are defined (non-uniquely) to minimize the expected set subject to the coverage $\mathcal{P}(g, \tau)$ being at least $1 - \alpha$:

$$g^\star, \tau^\star := \underset{g:\mathcal{X}_m \times \mathcal{Y}_m \mapsto \mathbb{R}, \tau \in \mathbb{R}}{\arg\min} S(g, \tau) \quad \text{s.t.} \quad \mathcal{P}(g, \tau) \geq 1 - \alpha. \quad \text{(P1)}$$

#### 3.1.2 PRACTICAL VERSION: DIFFERENTIABLE SURROGATES AND EMPIRICAL ESTIMATES

Problem (P1) characterizes optimal score functions and thresholds. However, in practice, we do not know the distribution and thus do not have access to the quantities in (4) and (5). Instead, we obtain their estimates using a finite training sample $D_{\text{train}} = \{(x_i, y_i^\star)\}_{i=1}^{n_t}$ drawn independently from the same distribution:

$$\widehat{S}(g, \tau) := \frac{1}{n_t} \sum_{i=1}^{n_t} \sum_{y \in \mathcal{Y}_m} \mathbb{1}\{g(x_i, y) \geq \tau\}, \quad \widehat{\mathcal{P}}(g, \tau) := \frac{1}{n_t} \sum_{i=1}^{n_t} \mathbb{1}\{g(x_i, y_i^\star) \geq \tau\}. \quad (6)$$

Using these plug-in estimators in problem (P1) yields a revised optimization problem. However, it is difficult to solve this problem as the objective and constraints are not differentiable. To make them differentiable, we introduce the following surrogates. Given $g(x, y)$ and $\tau$, define the following sigmoid function with $\beta > 0$, $\sigma(x, y, g, \tau, \beta) := 1/\big(1 + \exp(-\beta (g(x, y) - \tau))\big)$. The sigmoid function provides a differentiable approximation to the indicator variable for $g(x, y) \geq \tau$. The approximation is tighter with ~~higher~~ larger $\beta$ i.e., $\sigma(x, y, g, \tau, \beta) \to \mathbb{1}\{g(x, y) \geq \tau\}$ as $\beta \to \infty$, and $g(x, y) \geq \tau \iff \sigma(x, y, g, \tau) \geq 1/2$. By using these sigmoid surrogates in equation (6), we obtain the following smooth plugin estimates,

$$\widetilde{S}(g, \tau) := \frac{1}{n_t} \sum_{i=1}^{n_t} \sum_{y \in \mathcal{Y}_m} \sigma(x_i, y, g, \tau, \beta), \quad \widetilde{\mathcal{P}}(g, \tau) := \frac{1}{n_t} \sum_{i=1}^{n_t} \sigma(x_i, y_i^\star, g, \tau, \beta). \quad (7)$$

It is easy to see that by the strong law of larger numbers and properties of the sigmoid function, as $n_t, \beta \to \infty$, the surrogate average set size and coverage will converge almost surely to their population versions, i.e. $\widetilde{S}(g, \tau) \xrightarrow{a.s.} S(g, \tau)$ and $\widetilde{\mathcal{P}}(g, \tau) \xrightarrow{a.s.} \mathcal{P}(g, \tau)$. We replace the expected set size and marginal coverage by these smooth surrogates in (P1) and transform it into an unconstrained problem with the penalty term $\lambda > 0$. We also introduce $\ell_2$ regularization to encourage low norm solutions. We optimize the score function $g$ over a flexible space of functions $\mathcal{G}$, such as neural networks (NNs). The resulting problem (P2) is differentiable, and we solve it on a training dataset $D_{\text{train}} = \{x_i, y_i^\star\}_{i=1}^{n_t}$ using stochastic gradient descent.

$$\tilde{g}, \tilde{\tau} := \underset{g \in \mathcal{G}, \tau \in \mathbb{R}}{\arg \min} \widetilde{S}(g, \tau) + \lambda\big(\widetilde{\mathcal{P}}(g, \tau) - 1 + \alpha\big)^2 - \hat{\mathcal{C}}(g) + \lambda_1 \|g\|_2^2. \tag{P2}$$

Here, $\hat{\mathcal{C}}(g) := \frac{1}{n_t} \sum_{i=1}^{n_t} \log(g(x_i, y_i^*))$ is the cross entropy term included to encourage higher scores for correct predictions and the regularization term $\lambda_1 \|g\|_2^2$ is the squared norm over the parameters of $g$ to promote low norm solutions. Solving (P2) yields a score function $\tilde{g}$ and a threshold $\tilde{\tau}$. However, $\tilde{\tau}$ may be biased, since it is estimated on the same data as $\tilde{g}$. Following the split conformal procedure, we therefore estimate a new threshold $\hat{\tau}$ on a separate calibration dataset. In practice, we use 3-layer neural networks with `tanh` activation as $\mathcal{G}$ and use the LLM's logits and the penultimate layer's representations corresponding to the last token as input features to the $g$ network (see Appendix for details). Note that our framework is flexible and can work with any choice of features and function class for which the $\ell_2$ norm can be calculated.

## 3.2 CONFORMAL REVISION OF QUESTIONS (CROQ)

Here we consider how conformal prediction sets can be used for downstream purposes other than uncertainty quantification, namely for improving the final accuracy in MCQ type tasks. Our procedure involves re-prompting the LLM with the reduced answer options from a conformal prediction set. We describe our procedure and then discuss the connection to the Monty Hall problem. The steps are also illustrated with an example in Figure 1.

**Scores and Threshold for Conformal Prediction.** We first fix a score function $g$, which could be any arbitrary function but which here we restrict to either logits from the LLM or our CP-OPT scores (Section 3.1). We then run the split conformal procedure with coverage level $1 - \alpha$ for some $\alpha \in [0, 1]$ to estimate the threshold $\hat{\tau}$. CROQ then proceeds as follows.

**Step 1: Get Conformal Prediction Set.** Given a test instance $x$, we generate a first stage prediction set, $C(x; g, \hat{\tau})$. Per the coverage guarantee (Proposition 2.1), we expect that the true answer $y^\star \in C(x; g, \hat{\tau})$ with probability at least $1 - \alpha$.

**Step 2: Revise and Ask the LLM Again.** If the first stage prediction set $C(x; g, \hat{\tau})$ is empty or is of size 1 or size $m$ (the number of answer options), then we simply utilize the LLM's initial answer, as described in section 2.1, since the conformal procedure has yielded no additional information. Otherwise, we modify the prompt $x$ to $x' = (Q, O')$, where $O' = \{(K_j, V_j) : K_j \in C(x; g, \hat{\tau})\}$. The keys in $O'$ are changed so that they start with the first letter of the alphabet and go to the letter corresponding to the number of choices available. For example, if there were initially four answer options $\{a, b, c, d\}$, and the conformal prediction set was $\{c, d\}$, then the two options in the set would receive new keys $\{a, b\}$. Then $x'$ is transformed into a prompt format and input to the LLM and the standard inference procedure (section 2.1) is run to extract the predicted answer key $\hat{y}'$.

Next, we discuss an interesting analogy between CROQ and the Monty Hall problem to provide insights into how CROQ can improve LLM accuracy and show its flexibility to accommodate oracles such as other LLMs.

**Connection to the Monty Hall Problem.** Monty Hall is a probability puzzle (Selvin, 1975; Granberg, 1999; Rosenthal, 2008) based on a popular game show, where a contestant chooses one of three doors (choices), behind one of which is a car and the others, goats. The contestant chooses an initial door, which remains closed. After the host reveals a goat behind another door, the contestant is asked if they want to switch doors. Under typical assumptions, switching offers a higher chance of winning, i.e. selecting the door with a car behind it.

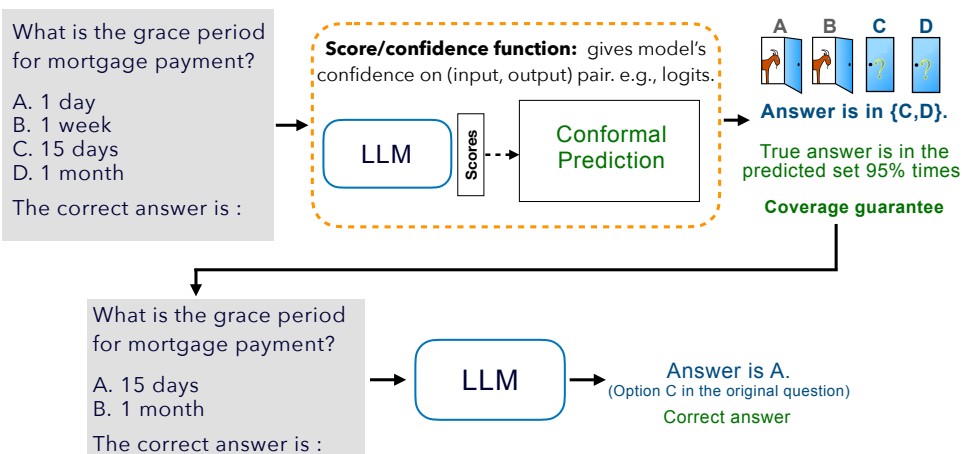

Figure 1: (CROQ) Illustration of conformal revision of questions and prompting the LLM with the revised question. In this example, the initial predicted set by LLM + conformal prediction (CP) is {C, D}. The question and labels are revised to contain only the answer choices in the prediction set and the LLM is prompted with the revised question. Since CP provides rigorous coverage guarantees, we expect that re-prompting the LLM with reduced answer choices will improve the chances of obtaining the correct answer. See Section 3.2 for more details.

In CROQ, we leverage conformal inference to eliminate "goats" (incorrect answers). The conformal set produced in the first stage of CROQ contains the correct answer with some user-specified probability, say 95%. In that sense, we imagine that an (imperfect or probabilistic) oracle is opening some number of doors (answer options) that with high probability reveal only goats (incorrect answers). Those answer options are eliminated, and then the LLM is queried again with the remaining set of answers. Thus, similar to the Monty Hall, there is a reduction in uncertainty after some doors (choices) are opened (eliminated), which we hypothesize leads to improved chances of winning (answering correctly). Hence, CROQ can enable an LLM agent to "self-correct".

Here the oracle's "knowledge" comes from the distribution of the scores of correct answers, so it is not contained within the LLM's representation of any given query. In that sense, it is extra information that gets added to the given query. Additionally, *the score function and the resulting distribution of scores can come from any source*. For example, from another LLM, from embeddings that measure semantic similarity between questions and answers, etc. The extra information that the conformal procedure represents can therefore come entirely from an external source.

**Coverage-Accuracy Trade-offs.** Conformal prediction has a small risk, namely a probability $\alpha$, of eliminating the door with the "car" (true choice). If $\alpha$ is too small, we will have high coverage but may have relatively large sets i.e. little reduction in the uncertainty. If $\alpha$ is too large, we will too frequently exclude the correct answer from the revised question. Since our interest is in improving final accuracy, we treat $\alpha$ as a tuning parameter that can be optimized for accuracy (Section 4.2).

# 4 EXPERIMENTS

We conduct experiments on benchmark MCQ and tool usage tasks with open-weight instruction-tuned models to test they following hypotheses:

**H1.** Using our CP-OPT scores in conformal prediction on MCQ tasks with LLMs yields a smaller average set size at the same level of coverage in comparison to using LLM logits. **H2.** Conformal revision of questions (CROQ) improves accuracy over the standard inference procedure. **H3.** CROQ with CP-OPT scores performs better than CROQ with logit scores.

## 4.1 EXPERIMENTAL SETUP

We first describe the setup for the experiments and then discuss the results for the above hypotheses.

**Datasets.** We evaluate our hypotheses on 3 datasets: MMLU (Hendrycks et al., 2021), TruthfulQA (Lin et al., 2022), and ToolAlpaca (Tang et al., 2023). MMLU and TruthfulQA are popular

| Model | # Opt. | MMLU | | | | ToolAlpaca | | | | TruthfulQA | | | |
|---|---|---|---|---|---|---|---|---|---|---|---|---|---|
| | | Avg. Set Size | | Coverage | | Avg. Set Size | | Coverage | | Avg. Set Size | | Coverage | |
| | | Logits | Ours | Logits | Ours | Logits | Ours | Logits | Ours | Logits | Ours | Logits | Ours |
| Llama-3 | 4 | 2.56 | **2.51*** | 95.81* | 95.35 | 1.17 | **1.16** | 96.38 | 96.03 | 3.22 | **2.71*** | 95.7* | 92.9 |
| | 10 | 5.19 | **4.73*** | 95.57* | 95.02 | 1.52 | **1.50** | 96.03 | 96.50 | 7.28 | **6.51*** | 93.7 | 93.7 |
| | 15 | 7.66 | **6.62*** | 95.36* | 94.60 | 2.25 | **1.66*** | 97.31* | 95.79 | 9.95 | **9.73*** | 94.7 | 92.4 |
| Phi-3 | 4 | 2.21 | **2.15*** | 94.6 | 94.6 | 1.11 | **1.06*** | 96.50 | 95.79 | 2.91 | **2.45*** | 97.0 | 96.5 |
| | 10 | 4.61 | **4.50*** | 94.7 | 94.5 | 1.27 | 1.27 | 95.33 | 95.68 | 6.90 | **6.34*** | 95.7 | 95.2 |
| | 15 | **6.46*** | 6.66 | 93.9 | 94.0 | 1.58 | **1.57** | 96.73 | 97.31 | 10.85 | **9.72*** | 95.9 | 95.9 |

Table 1: Average set sizes and coverage rates (in percentages) for conformal prediction sets on the MMLU, ToolAlpaca, and TruthfulQA datasets using `Llama-3-8B-Instruct` (Llama-3) and `Phi-3-4k-mini-Instruct` (Phi-3), with a target coverage level of $95\%$. For each dataset, we vary the number of answer options. Using CP-OPT for the score function produces smaller average set sizes more frequently compared to using logits. Bold numbers indicate smaller average set sizes. Shaded cells indicate settings where CP-OPT results in smaller set sizes and equal or larger coverage. Asterisks on the larger of a pair of numbers indicate where the difference in average set size or coverage is statistically significant at the $\alpha = 0.05$ level. See Appendix C for details.

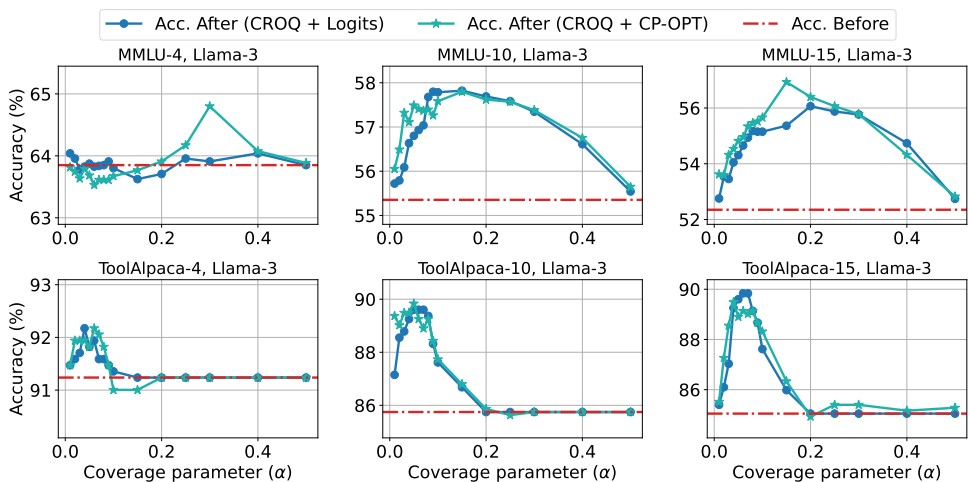

Figure 2: Accuracy on revised questions on the MMLU and ToolAlpaca dataset while varying coverage parameter $\alpha$ for `Llama-3-8B-Instruct` (Llama-3) model and both scores. Smaller values of $\alpha$ correspond to high levels of coverage. When coverage is too large, few or no answers are eliminated, and the LLM is prompted with the same question. When coverage is low, a larger portion of answer sets no longer contain the true answer or produce empty prediction sets thus resulting in diminished benefits of revision.

benchmark datasets for multiple choice questions. MMLU focuses on assessing multitask accuracy, containing multiple choice questions (MCQs) from 57 domains, including humanities, math, medicine, etc.; while TruthfulQA evaluates an LLM's ability to answer truthfully and not mimic preconceived falsehoods that humans are susceptible to. ToolAlpaca contains 3.9k tool-use instances from a multi-agent simulation environment, which we augment to a MCQ format. Dataset descriptions, and example questions and responses, are provided in Appendix D.

**Models.** We use auto-regressive language models based on the transformer architecture. We choose instruction-tuned, open-weight, and small to medium sized models, for reproducibility and reduced computational cost. Specifically, we use `Llama-3-8B-Instruct` (Dubey et al., 2024) by Meta and `Phi-3-4k-mini-Instruct` (Abdin et al., 2024) by Microsoft. We provide results on `gemma-2-9b-it-SimPO` model (Meng et al., 2024) in the Appendix B.2 (Tables 11, 12, and 13).

**Choices of Scores.** We use the following scores for conformal prediction. (1) LLM Logits (`Softmax`) are extracted from the LLM as discussed in Section 2.1. These have been used in prior

| Model | Score | Set Size | 1 | 2 | 3 | 4 | 5 | 6 | Overall |
|-------|-------|----------|-----|-----|-----|-----|-----|-----|---------|
| Llama-3 | Logits | **Coverage** | 95.27 | 97.03 | 97.10 | 100.00 | 100.00 | 100.00 | 96.03 |
| | | **Fraction** | 59.23 | 31.43 | 8.06 | 0.93 | 0.23 | 0.12 | 100.00 |
| | | **Acc. Before** | 95.27 | 77.32 | 53.62 | 37.50 | 100.00 | 100.00 | 85.75 |
| | | **Acc. After** | 95.27 | **84.39** | **71.01** | **62.50** | 100.00 | 100.00 | **89.60*** |
| | Ours | **Coverage** | 95.77 | 94.97 | 100.00 | 0.00 | 0.00 | 0.00 | 96.50 |
| | | **Fraction** | 55.84 | 38.32 | 5.61 | 0.23 | 0.00 | 0.00 | 100.00 |
| | | **Acc. Before** | 96.23 | 74.39 | 62.50 | 0.00 | 0.00 | 0.00 | 85.75 |
| | | **Acc. After** | 96.23 | **82.93** | **75.00** | **50.00** | 0.00 | 0.00 | **89.84*** |
| Phi-3 | Logits | **Coverage** | 95.77 | 94.97 | 100.00 | 0.00 | 0.00 | 0.00 | 95.33 |
| | | **Fraction** | 74.18 | 24.18 | 1.64 | 0.00 | 0.00 | 0.00 | 100.00 |
| | | **Acc. Before** | 95.12 | 64.73 | 42.86 | 0.00 | 0.00 | 0.00 | 86.92 |
| | | **Acc. After** | 95.12 | **78.74** | **71.43** | 0.00 | 0.00 | 0.00 | **90.77*** |
| | Ours | **Coverage** | 95.77 | 94.97 | 100.00 | 0.00 | 0.00 | 0.00 | 95.68 |
| | | **Fraction** | 74.65 | 23.25 | 2.10 | 0.00 | 0.00 | 0.00 | 100.00 |
| | | **Acc. Before** | 95.77 | 60.80 | 61.11 | 0.00 | 0.00 | 0.00 | 86.92 |
| | | **Acc. After** | 95.77 | **74.37** | **77.78** | 0.00 | 0.00 | 0.00 | **90.42*** |

Table 2: Results for CROQ experiment on ToolAlpaca dataset with 10 response options. Here we used $\alpha = 0.05$. The questions are binned according to the set sizes and we report the coverage, fraction of points, accuracy before CROQ and accuracy after CROQ in each bin. We see a consistent improvement in the overall accuracy by around 4% across all model and score combinations and about 14% on questions with set size 2, with Phi-3 model, amounting to about 24% of the questions.

works (Kumar et al., 2023; Su et al., 2024). (2) CP-OPT (Ours) are the scores learned using the score optimization procedure discussed in Section 3.1. We use the train split for each dataset to learn these scores. The hyperparameter settings we used for CP-OPT are given in Appendix D.3. We omit the self-consistency based heuristic scores proposed by Su et al. (2024), as these require repeated inferences to get good estimates of the scores, and hence have a high computational cost.

We use the provided validation splits as our calibration datasets for the conformal procedure. For testing hypothesis **H1**, where our interest is in obtaining small sets with high coverage, we calibrate the conformal threshold for the coverage guarantee of $95\%$, i.e. we set the error rate $\alpha$ to 0.05. The hyperparameters used to learn the score function using SGD are provided in table 22 in Appendix D.3. For testing **H2**, we calibrate to a range of $\alpha$ values: {0.01, 0.02, 0.03, 0.04, 0.05, 0.06, 0.07, 0.08, 0.09, 0.1, 0.15, 0.2, 0.25, 0.3, 0.4, 0.5 }. Performance is computed on test splits.

### 4.2 DISCUSSION

*H1. Improvement in conformal set sizes with our CP-OPT scores.* We run the CP procedure using the LLM logits and CP-OPT scores and obtain conformal sets for points in the test sets. We compute the average set size and coverage for each dataset, model, and score combination. The results are in Table 10. As expected, in most cases we see a drop in the set sizes with our scores, which is usually statistically significant. The decrease gets more pronounced with a higher number of options. In some cases, the reduction in set size is accompanied by a statistically significant reduction in coverage relative to using the logits as scores. (Since the target coverage level is $95\%$, anything above 95 is over-coverage, while anything below 95 is under-coverage.) We cannot tell from these results to what extent the reduction in coverage accounts for the reduction in set sizes; we leave such an investigation for future work. However, in five settings, highlighted in gray, the CP-OPT scores result in smaller sets and equal or larger coverage, while the opposite never happens.

*H2. Accuracy improvement with conformal revision of questions (CROQ).* Figure 2 shows the accuracy before and after CROQ for a range of $\alpha$ values with Llama-3. All three datasets show improvements in accuracy for appropriate values of $\alpha$. In Table 2, we see that accuracy improvements also occur conditional on prediction set sizes, with some set sizes corresponding to relatively large improvements. We additionally see that accuracy declines approximately monotonically as a function of set size, indicating that set size is a good measure of the LLM's uncertainty. Comprehensive additional results for the CROQ experiments are given in Appendix B.2. This suggests a procedure in which set size is used to decide whether to defer to a human.

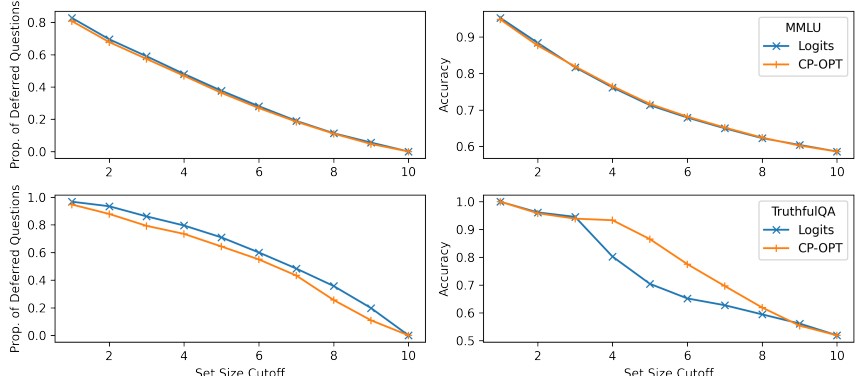

Figure 3: Proportion of questions deferred to a human when conformal prediction set sizes exceed a certain cutoff, and the corresponding LLM accuracy for questions (without revision) retained by the LLM as a function of cutoff threshold. In the top row (MMLU, 10 options), the difference in deferral and accuracy for is negligible, whereas in the bottom row (TruthfulQA, 10 options), CP-OPT defers fewer questions to the human while providing similar or improved accuracy for questions retained.

*H3. CROQ with CP-OPT scores is better than CROQ with logit scores.* CP-OPT scores are designed to minimize set sizes while maintaining the coverage guarantee. As a result, incorporating these scores into the CROQ procedure is expected to significantly reduce uncertainty for many questions, leading to fewer answer options in the revised prompts. We expect LLMs are more likely to answer correctly when prompted with the revised question with fewer options. The results presented in Tables 13, 12, 11, 18, and 19 align with this expectation, showing that CROQ with CP-OPT scores achieves noticeably higher accuracy compared to CROQ with logits. However, in some settings (e.g., Tables 9, 5, 4, 6, 7, 15, and 17) CROQ with CP-OPT has similar or minutely better overall accuracy than CROQ with logits. A plausible explanation is that the reduction in set size by CP-OPT was not substantial enough to result in significant improvements. Nonetheless, the overall results highlight that CROQ with CP-OPT is generally better than CROQ with logits.

Figure 3 illustrates a procedure in which a set size cutoff is selected, and the LLM answer is only retained if the set size is at or below that cutoff. For all larger sets, the question is passed to a human. As desired, lower set size cutoffs result in higher accuracy. As the set size cutoff increases, the accuracy approaches the LLM's marginal accuracy, while the cost of deferral (i.e. the cost of having a human rather than an LLM answer the question) decreases. In the top row, we see a setting (which is representative of our results) in which there is no substantial difference between logits and the CP-OPT scores. In the bottom row, we see a setting in which the CP-OPT scores result in both higher accuracy and lower deferral cost for most set size cutoffs.

## 5   RELATED WORK

**Conformal Prediction for Uncertainty Quantification with LLMs**   Recently there has been growing interest in using conformal prediction to quantify and control uncertainty in LLM-related tasks. In the context of multi-choice question answering (MCQ), previous works have investigated a variety of conformal score functions, including (the softmax of) the LLM logits corresponding to the response options (Kumar et al., 2023; Ren et al., 2023) or functions thereof (Ye et al., 2024), confidence scores generated by the LLM itself, or "self-consistency" scores derived by repeated querying of the LLM (Su et al., 2024). We build on this work by aiming to learn a conformal score function that yields small conformal sets, rather than taking the score function as given.

In addition to the MCQ setting, there has been recent work utilizing conformal prediction in the context of open-ended response generation (Quach et al., 2024; Mohri and Hashimoto, 2024; Cherian et al., 2024). This setting differs in that there is not necessarily a unique correct response, so the notion of coverage must be redefined around *acceptability* or *factuality* rather than correctness. When factuality is the target, the goal is to calibrate a pruning procedure that removes a minimal number of claims from an LLM-generated open response, such that the remaining claims are all factual with high probability; that is, the goal is to retain as large a set as possible, rather than to generate a set

with the smallest number of responses possible as in MCQ. Conformal prediction has also been used to capture token-level uncertainty (Deutschmann et al., 2024; Ulmer et al., 2024).

**Optimizing conformal prediction procedures** Several recent works have considered how to learn good conformal score functions from data, primarily in the context of supervised learning models (Bai et al., 2022; Stutz et al., 2022; Yang and Kuchibhotla, 2024; Xie et al., 2024). With LLMs, Cherian et al. (2024) consider how to learn a good score function to achieve factuality guarantees; their optimization problem differs from ours due to the difference in setting as well as the addition of conditional coverage constraints (ensuring that coverage holds in different parts of the feature space). Kiyani et al. (2024) design a framework to minimize the size ("length," in their terminology) of conformal sets, which they apply to MCQ as well as to supervised learning problems. However, their framework is concerned with how to generate sets given a model and a conformity score, rather than how to learn a conformity score.

The works mentioned above all aim to produce (small) conformal sets that satisfy coverage guarantees. Among these, only Ren et al. (2023) consider how conformal sets may be used downstream, in their case to improve the efficiency and autonomy of robot behavior. To our knowledge, our work is the first to investigate whether conformal prediction can be used to increase the accuracy of LLMs on MCQ type tasks.

# 6 CONCLUSION AND FUTURE WORK

We investigated how conformal prediction can be used to quantify and reduce output uncertainty for decision-making problems such as tool selection and multi-choice question answering (MCQ) with LLMs. We defined an optimal conformal score function (P1) that minimizes average set size subject to a coverage constraint, and we showed how to estimate it using a differentiable loss function that can be optimized via stochastic gradient descent (P2). We called this procedure CP-OPT. In experiments with a variety of models, datasets, and answer option cardinalities, we showed that CP-OPT results in smaller average set sizes than the baseline score function consisting of the LLM logits corresponding to the MCQ answer options, although these smaller set sizes sometimes resulted in reduced coverage. We emphasize that CP-OPT is extremely general and can be applied with different models, features sets, etc. If the LLM logits are already highly informative with regard to model uncertainty, then there may be little to gain (but nothing to lose) from applying CP-OPT, but in other settings there may be substantial benefit from optimizing the conformal scores.

We further investigated whether re-prompting an LLM with the answer options contained in the conformal set would result in higher final accuracy on MCQ tasks. We called this procedure CROQ. The intuition is that conformal sets will contain the true answer with high probability while potentially substantially reducing the number of answer options the LLM has to consider. We found that CROQ increases accuracy in most of the cases and that there is an interplay between the coverage level and the accuracy improvement which can be optimized. As an additional consideration of how prediction sets can be used for purposes other than uncertainty quantification, we illustrated how to use them to decide when to defer to human judgment, with both the accuracy and the deferral cost increasing as the set size cutoff shrinks.

A particularly interesting extension to the CROQ procedure is performing multiple rounds of question revisions. While we have demonstrated that a reduction in the number of available response options increases LLM accuracy, we conjecture that a further reduction in conformal set sizes could occur as well. Repeated conformal inference would come at higher computational costs, as performing it at each round would require repeated calibration, and may require fresh data to calibrate on. Additionally, without adjusting the coverage level used at each round, the final coverage rate will be lower than the nominal rate. Developing methodology to make a multi-round CROQ procedure both efficient and technically sound is a promising line of future research. Different models could also be used to generate the initial prediction set and the final answer, instead of a single LLM for both.

Other future lines of investigation include considering how to calibrate conformal score thresholds when the number of answer options that the LLM may encounter in MCQ may vary. This is relevant for example in tool usage problems, where an LLM may be asked to consider very different numbers of APIs for different types of queries. One possible approach involves quantile regression against the numbers of answer options, which may enable reasonable estimation of the quantiles in cases where some numbers of answer options are not well represented in the calibration data.

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

# A  DETAILS ON LLM INFERENCE IN MULTI-CHOICE QUESTION ANSWERING

We provide a formal description of the inference procedure described in the **LLM Inference** paragraph of Section 2.1.

The input prompt $x$ is a sequence of tokens $t_1, t_2, \ldots t_n$. We run the forward pass of the auto-regressive LLM (Touvron et al., 2023; Dubey et al., 2024; Abdin et al., 2024) on $x$ to produce a set of output logits:

$$\boldsymbol{l}_1, \boldsymbol{l}_2, \ldots, \boldsymbol{l}_n \leftarrow \texttt{LLM}\big(t_1, t_2, \ldots t_n\big) \tag{8}$$

Here, each logit $\boldsymbol{l}_j \in \mathbb{R}^{|V|}$ expresses the likelihood of the next token after $t_1, \ldots, t_j$, where $V$ is the universal set of tokens (aka the alphabet) for the given LLM and $|V|$ is its size. The last token's logits $\boldsymbol{l}_n$ are expected to have a high value for the correct answer key. We extract the logit vector $\bar{\boldsymbol{l}} \in \mathbb{R}^m$ corresponding to the option keys as follows:

$$\bar{\boldsymbol{l}} := \big[\ \boldsymbol{l}_n[Y_1],\ \boldsymbol{l}_n[Y_2],\ \ldots, \boldsymbol{l}_n[Y_m]\ \big], \tag{9}$$

where $\boldsymbol{l}_n[Y_j]$ denotes the logit value corresponding to the token $Y_j$ in the last token's logits $\boldsymbol{l}_n$. The logits $\bar{\boldsymbol{l}}$ are converted to softmax scores $s(x)$. The softmax score of point $x$ and option key $y$ is denoted by $s(x, y)$ and the predicted answer key $\hat{y}$ corresponds to the maximum softmax value:

$$s(x) := \texttt{softmax}(\bar{\boldsymbol{l}}), \qquad s(x, y) := s(x)[y], \qquad \hat{y} := \underset{y \in \{Y_1, \ldots Y_m\}}{\arg\max}\ s(x, y) \tag{10}$$

# B  ADDITIONAL DETAILS RESULTS

This appendix contains additional results and details not included in the main paper due to length constraints.

## B.1  DETAILS OF FEATURES AND $\mathcal{G}$ USED IN EXPERIMENTS

Let $\boldsymbol{z} \in \mathbb{R}^{d+m}$ be the concatenation of the LLM's penultimate layer's representation ($d$-dimensional) and logits ($m$-dimensional) for the last token. Our choice of $\mathcal{G}$ for the experiments is defined as follows,

$$\mathcal{G} := \{g : \mathbb{R}^{d_0} \to \Delta^{m-1} \mid g(\boldsymbol{z}) := \texttt{softmax}(\boldsymbol{W}_3 \texttt{tanh}(\boldsymbol{W}_2 \texttt{tanh}(\boldsymbol{W}_1(\boldsymbol{z})))),$$
$$\boldsymbol{W}_1 \in \mathbb{R}^{d_0 \times d_1}, \boldsymbol{W}_2 \in \mathbb{R}^{d_1 \times d_2}, \boldsymbol{W}_3 \in \mathbb{R}^{d_2 \times m}\}$$

Here, $d_0 = d + m, d_1 = (d + m)/2$, and $d_3 = (d + m)/4$ and $\Delta^{m-1}$ is the $m - 1$ dimensional probability simplex.

## B.2  RESULTS

Figure 5 shows accuracy after the CROQ procedure as a function of $\alpha$ for Phi-3. The results are qualitatively similar to the results for Llama-3 in the main text (Section 4.2).

Table 3 illustrates how often CROQ causes the LLM's answer to change relative to the baseline inference procedure. In Monty Hall terms (Section 3.2), these represent how often the LLM "switches" its answer in response to the elimination of some answer options. We see that the LLM switches some initially correct answers to wrong answers and vice versa but that the *wrong-to-right* switch frequency generally outweighs the *right-to-wrong* switch frequency, resulting in higher accuracy after CROQ.

Figure 4 provides motivation for CROQ, demonstrating that a reduction of set size leads to improved accuracy. Using the Truthful QA dataset with 15 response options, we first construct conformal prediction sets using logits. With these prediction sets, we then leverage oracle knowledge to reduce the prediction set size by 0 to 10 options - with 0 eliminations being the regular CROQ method. Incorrect answers are randomly eliminated, ensuring that coverage remains constant. As more incorrect answers are eliminated, the accuracy of the LLM after requerying increases as more answers are removed (smaller prediction set). These results motivate the use of a score function which minimizes set size while controlling coverage.

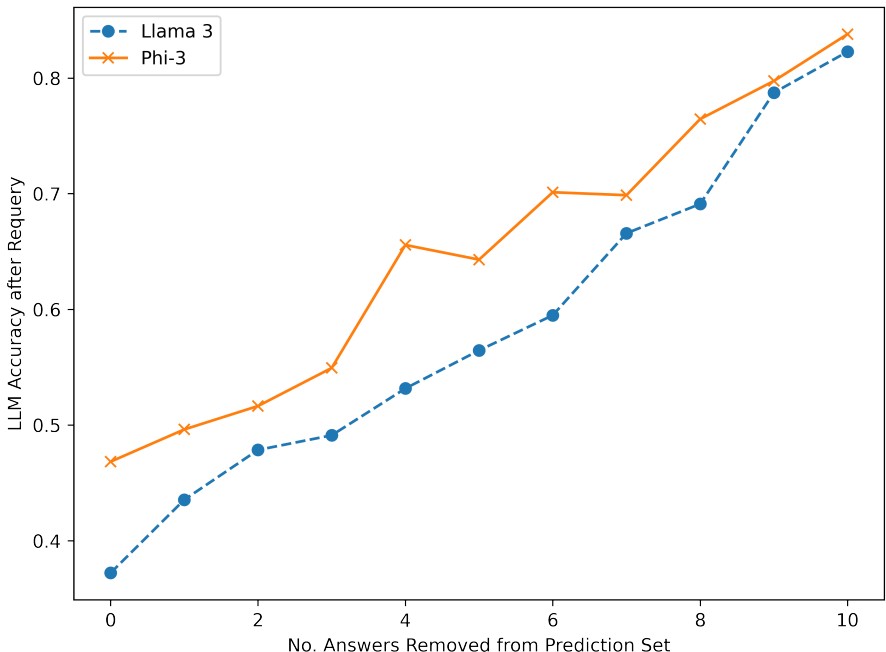

Figure 4: Requerying Accuracy of LLMs following oracle reduction of conformal prediction set size for the TruthfulQA dataset with 15 response options. As more answers are eliminated, set sizes become smaller, and the accuracy of the LLM following requerying increases.

All remaining results are organized by dataset. Tables for the CROQ results which illustrate accuracy changes conditional on set size are based on a confidence level of $95\%$ (equivalently an $\alpha$ level of 0.05). Note that with the ToolAlpaca dataset, not all possible set sizes occur, in which case we omit the corresponding columns. For example, with 10 response options, only sets of size 8 and smaller occur.

Asterisks in the tables indicate where the difference in overall accuracy from Before to After, i.e. from baseline to after the CROQ procedure, is statistically significant at the $\alpha = 0.05$ level. (In some tables, like Table 8, none of the changes are significant.) See Appendix C for details on how statistical significance was calculated.

### B.3 MMLU

Results for the experiments on the MMLU dataset are given in Tables 5 to 9 and Figures 6 to 8.

### B.4 TRUTHFULQA

Results for the experiments on the TruthfulQA dataset are given in Tables 14 to 19 and Figures 12 and 13.

### B.5 TOOLALPACA

Results for experiments on the ToolAlpaca dataset are given in Tables 2, 20 and 21 and Figures 9 and 10.

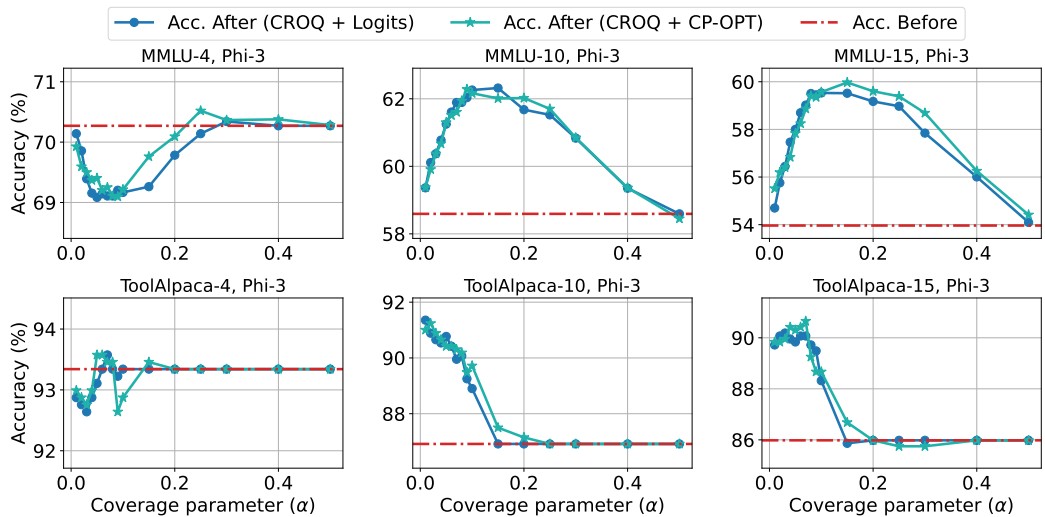

Figure 5: Accuracy on revised questions on the MMLU and ToolAlpaca dataset while varying coverage parameter $\alpha$ for `Phi-3-4k-mini-Instruct` (Phi-3) model and both scores. Smaller values of $\alpha$ correspond to high levels of coverage. When coverage is too large, few or no answers are eliminated, and the LLM is prompted with the same question. When coverage is low, a larger portion of answer sets no longer contain the true answer and the benefits of revision are diminished.

| Model | # Opt. | MMLU | | | | ToolAlpaca | | | | TruthfulQA | | | |
| | | Change R→W | | Change W→R | | Change R→W | | Change W→R | | Change R→W | | Change W→R | |
| | | Logits | Ours | Logits | Ours | Logits | Ours | Logits | Ours | Logits | Ours | Logits | Ours |
|---|---|---|---|---|---|---|---|---|---|---|---|---|---|
| Llama-3 | 4 | **1.70** | 2.10 | 1.80 | **1.90** | 1.40 | **1.10** | **2.00** | 1.60 | **0.80** | 2.00 | 2.00 | **5.30** |
| | 10 | **4.20** | 5.40 | 5.70 | **7.60** | 2.50 | 2.70 | 6.30 | **6.80** | 2.80 | 3.00 | 3.50 | **6.60** |
| | 15 | **4.70** | 6.60 | 6.60 | **9.10** | 2.80 | **2.50** | 7.40 | 6.30 | 5.10 | 6.10 | 4.30 | **6.30** |
| Phi-3 | 4 | 3.90 | **3.40** | **2.70** | 2.50 | 1.60 | **1.10** | **1.40** | 1.30 | **1.00** | 1.50 | 1.30 | **2.00** |
| | 10 | **5.00** | 5.40 | 7.70 | **8.10** | 1.90 | 2.20 | **5.70** | 5.70 | 2.30 | 3.80 | 3.50 | **6.10** |
| | 15 | **5.10** | 5.60 | 9.20 | **9.50** | 3.90 | 3.90 | 7.70 | **8.20** | 2.50 | 2.50 | 2.30 | **5.60** |

Table 3: Percentage of answers that change as a result of the CROQ procedure. R→W indicates questions that are answered correctly ("Right") before CROQ and incorrectly ("Wrong") after CROQ, while W→R indicates the opposite. CROQ results in changes to the final answer in roughly 3-15% of questions, with larger numbers of response options resulting in more frequent changes. Bold numbers indicating the larger number for logits vs. our scores.

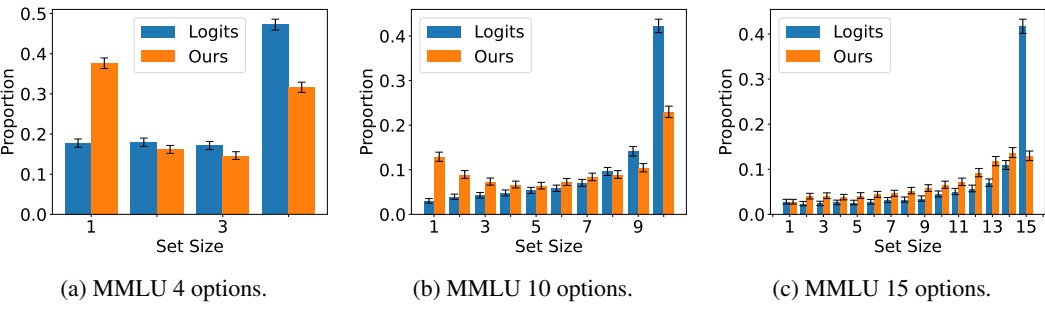

(a) MMLU 4 options.  (b) MMLU 10 options.  (c) MMLU 15 options.

Figure 6: Distributions of sizes of sets obtained from CP-OPT and logit scores on MMLU dataset and Gemma-2 model setting.

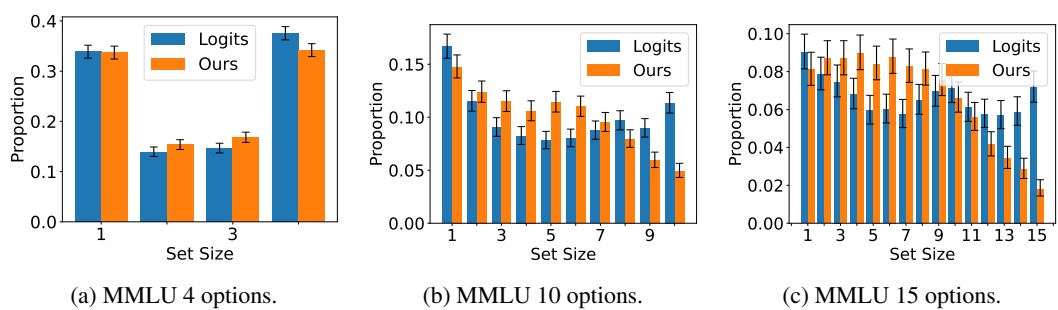

(a) MMLU 4 options.    (b) MMLU 10 options.    (c) MMLU 15 options.

Figure 7: Distributions of sizes of sets obtained from CP-OPT and logit scores on MMLU dataset and Llama-3 model setting.

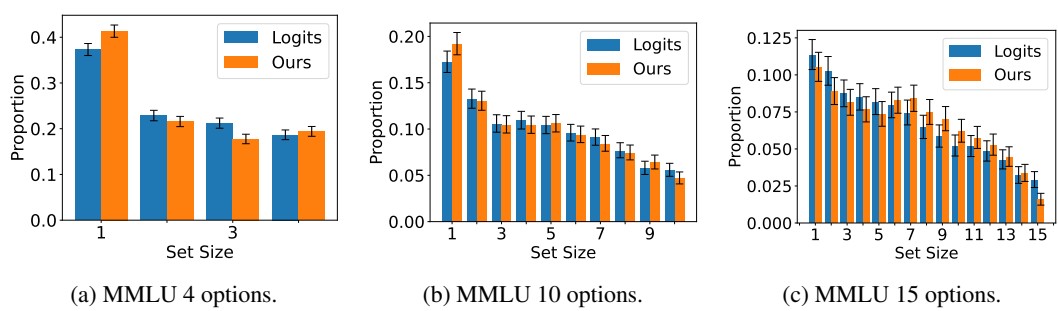

(a) MMLU 4 options.    (b) MMLU 10 options.    (c) MMLU 15 options.

Figure 8: Distributions of sizes of sets obtained from CP-OPT and logit scores on MMLU dataset and Phi-3 model setting.

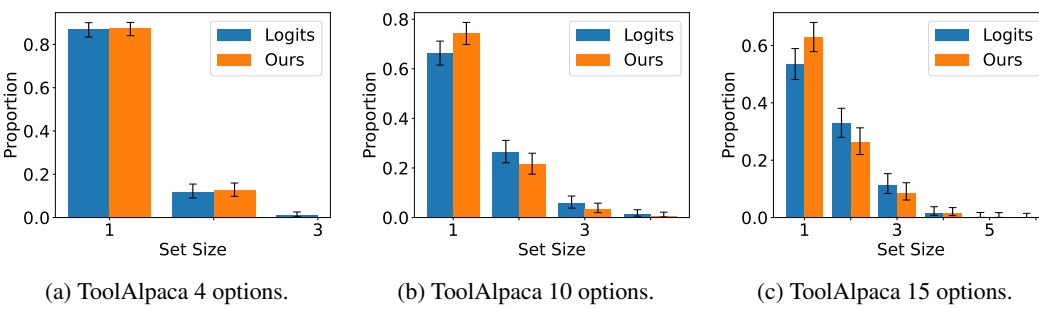

(a) ToolAlpaca 4 options.   (b) ToolAlpaca 10 options.   (c) ToolAlpaca 15 options.

Figure 9: Distributions of sizes of sets obtained from CP-OPT and logit scores on ToolAlpaca dataset and Llama-3 model setting.

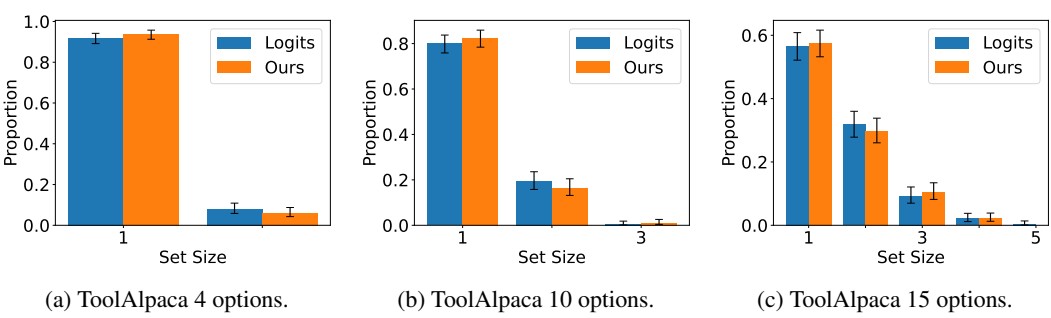

(a) ToolAlpaca 4 options.   (b) ToolAlpaca 10 options.   (c) ToolAlpaca 15 options.

Figure 10: Distributions of sizes of sets obtained from CP-OPT and logit scores on ToolAlpaca dataset and Phi-3 model setting.

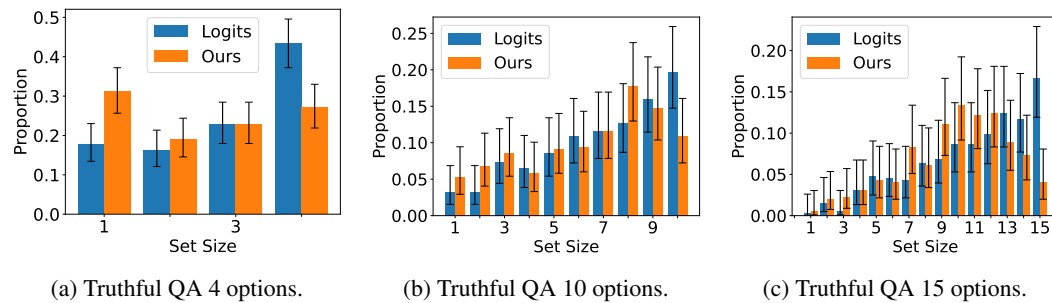

(a) Truthful QA 4 options.  (b) Truthful QA 10 options.  (c) Truthful QA 15 options.

Figure 11: Distributions of sizes of sets obtained from CP-OPT and logit scores on Truthful QA dataset and Phi-3 model setting.

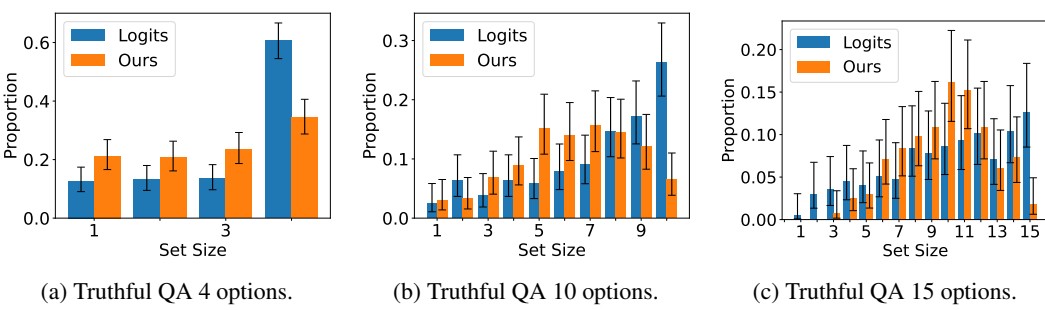

(a) Truthful QA 4 options.  (b) Truthful QA 10 options.  (c) Truthful QA 15 options.

Figure 12: Distributions of sizes of sets obtained from CP-OPT and logit scores on Truthful QA dataset and Llama-3 model setting.

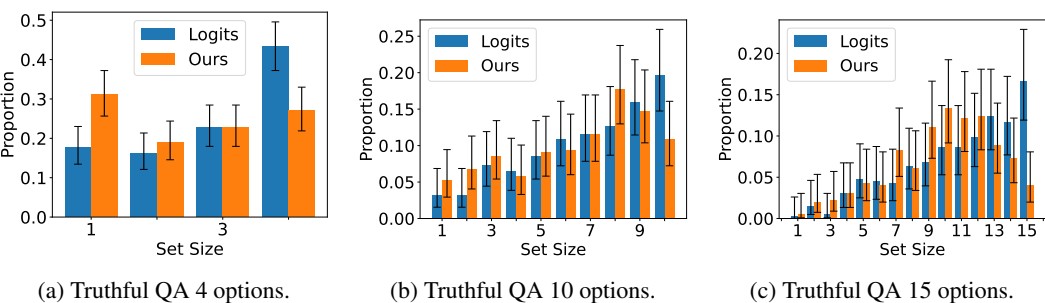

(a) Truthful QA 4 options.  (b) Truthful QA 10 options.  (c) Truthful QA 15 options.

Figure 13: Distributions of sizes of sets obtained from CP-OPT and logit scores on Truthful QA dataset and Phi-3 model setting.

| Set Size | 1 | 2 | 3 | 4 | 5 | 6 | 7 | 8 | 9 | 10 | 11 | 12 | 13 | 14 | 15 | Overall |
|---|---|---|---|---|---|---|---|---|---|---|---|---|---|---|---|---|
| Coverage | 94.14 | 93.17 | 91.12 | 93.39 | 93.37 | 94.72 | 94.26 | 95.76 | 95.75 | 94.43 | 96.82 | 97.13 | 96.89 | 98.75 | 100.00 | 94.60 |
| Fraction | 8.11 | 8.69 | 8.69 | 8.97 | 8.41 | 8.77 | 8.27 | 8.12 | 7.54 | 6.61 | 5.59 | 4.14 | 3.43 | 2.85 | 1.82 | 100.00 |
| Acc. Before | 94.14 | 84.02 | 71.04 | 63.76 | 54.44 | 50.34 | 41.32 | 41.52 | 35.59 | 34.11 | **29.30** | 28.65 | **29.07** | 21.25 | 20.92 | 52.35 |
| Acc. After | 94.14 | **84.29** | **73.77** | **67.20** | **56.28** | **54.40** | **48.49** | **44.44** | **39.06** | **36.62** | 28.87 | **32.09** | 27.68 | 20.83 | 20.92 | **54.82\*** |

Table 4: Results for CROQ experiment on MMLU dataset with 15 response options, using `Llama-3-8B-Instruct` and CP-OPT scores.

| Set Size | 1 | 2 | 3 | 4 | 5 | 6 | 7 | 8 | 9 | 10 | 11 | 12 | 13 | 14 | 15 | Overall |
|---|---|---|---|---|---|---|---|---|---|---|---|---|---|---|---|---|
| Coverage | 95.26 | 91.54 | 92.37 | 93.37 | 92.81 | 93.87 | 95.25 | 95.06 | 95.73 | 95.18 | 97.86 | 96.70 | 98.12 | 98.79 | 100.00 | 95.36 |
| Fraction | 9.02 | 7.86 | 7.46 | 6.80 | 5.95 | 6.01 | 5.74 | 6.49 | 6.94 | 7.14 | 6.10 | 5.76 | 5.68 | 5.87 | 7.17 | 100.00 |
| Acc. Before | 95.26 | 83.38 | 73.45 | 65.27 | 58.88 | 52.77 | 44.42 | 42.78 | 44.27 | **35.22** | 39.49 | 32.37 | 33.19 | 27.88 | **26.49** | 52.35 |
| Acc. After | 95.26 | **83.99** | **77.11** | **70.86** | **61.08** | **58.10** | **47.73** | **46.80** | **46.84** | 34.88 | **40.66** | **33.40** | **33.61** | 27.88 | 26.49 | **54.31\*** |

Table 5: Results for CROQ experiment on MMLU dataset with 15 response options, using `Llama-3-8B-Instruct` and logits.

| Set Size | 1 | 2 | 3 | 4 | 5 | 6 | 7 | 8 | 9 | 10 | 11 | 12 | 13 | 14 | 15 | Overall |
|---|---|---|---|---|---|---|---|---|---|---|---|---|---|---|---|---|
| Coverage | 95.81 | 93.16 | 91.01 | 91.88 | 93.16 | 93.11 | 93.76 | 94.11 | 93.69 | 94.74 | 96.08 | 95.81 | 93.58 | 95.91 | 100.00 | 93.96 |
| Fraction | 11.33 | 10.23 | 8.71 | 8.47 | 8.15 | 7.93 | 7.42 | 6.44 | 5.83 | 5.19 | 5.15 | 4.82 | 4.25 | 3.19 | 2.88 | 100.00 |
| Acc. Before | 95.81 | 81.55 | 67.30 | 58.40 | 53.86 | 50.45 | 41.76 | 40.52 | 40.12 | 38.67 | 35.02 | 28.82 | **24.30** | 21.56 | 20.58 | 53.96 |
| Acc. After | 95.81 | **85.03** | **73.84** | **66.53** | **62.30** | **54.64** | **48.80** | **44.01** | **44.60** | **42.11** | **37.79** | **30.54** | 23.18 | **22.68** | 20.58 | **58.00\*** |

Table 6: Results for CROQ experiment on MMLU dataset with 15 response options, using `Phi-3-4k-mini-Instruct` and logits.

| Set Size | 1 | 2 | 3 | 4 | 5 | 6 | 7 | 8 | 9 | 10 | 11 | 12 | 13 | 14 | 15 | Overall |
|---|---|---|---|---|---|---|---|---|---|---|---|---|---|---|---|---|
| Coverage | 95.25 | 92.64 | 92.09 | 92.69 | 92.22 | 93.96 | 94.05 | 95.06 | 93.05 | 96.35 | 94.20 | 94.80 | 93.58 | 97.15 | 100.00 | 94.02 |
| Fraction | 10.50 | 8.87 | 8.11 | 7.63 | 7.32 | 8.25 | 8.38 | 7.45 | 7.00 | 6.18 | 5.73 | 5.25 | 4.44 | 3.33 | 1.55 | 100.00 |
| Acc. Before | 95.25 | 85.27 | 73.65 | 66.10 | 56.24 | 52.37 | 46.18 | 38.54 | 36.44 | 34.36 | 33.75 | 30.77 | 23.53 | 20.28 | 16.79 | 53.96 |
| Acc. After | 95.25 | **86.08** | **78.18** | **72.94** | **62.40** | **59.28** | **53.40** | **46.50** | **42.03** | **35.70** | **36.02** | **31.67** | 23.53 | **21.35** | 16.79 | **57.83\*** |

Table 7: Results for CROQ experiment on MMLU dataset with 15 response options, using `Phi-3-4k-mini-Instruct` and CP-OPT.

| Model | Score | Set Size | 1 | 2 | 3 | 4 | Overall |
|---|---|---|---|---|---|---|---|
| Llama-3 | Logits | Coverage | 93.58 | 92.8 | 93.03 | 100.00 | 95.81 |
| | | Fraction | 33.87 | 13.93 | 14.66 | 37.54 | 100.00 |
| | | Acc. Before | 93.58 | 70.18 | **49.39** | 40.30 | **63.84** |
| | | Acc. After | 93.58 | **70.44** | 48.09 | 40.30 | 63.69 |
| | Ours | Coverage | 93.55 | 90.71 | 93.71 | 100.00 | 95.37 |
| | | Fraction | 33.68 | 15.35 | 16.81 | 34.17 | 100.0 |
| | | Acc. Before | 93.55 | **69.14** | **53.53** | 37.27 | **63.84** |
| | | Acc. After | 93.55 | 68.14 | 52.47 | 37.27 | 63.52 |
| Phi-3 | Logits | Coverage | 94.75 | 91.48 | 93.17 | 100.0 | 94.64 |
| | | Fraction | 37.30 | 22.86 | 21.20 | 18.64 | 100.0 |
| | | Acc. Before | 94.75 | **70.25** | **52.68** | 41.31 | **70.27** |
| | | Acc. After | 94.75 | 66.92 | 50.67 | 41.31 | 69.08 |
| | Ours | Coverage | 93.88 | 91.55 | 94.37 | 100.0 | 94.64 |
| | | Fraction | 41.32 | 21.55 | 17.73 | 19.39 | 100.0 |
| | | Acc. Before | 93.88 | **67.78** | **52.14** | 39.29 | **70.27** |
| | | Acc. After | 93.88 | 64.81 | 50.87 | 39.29 | 69.41 |

Table 8: Results for CROQ experiment on MMLU dataset with 4 response options. In this setting, the CROQ procedure leads to a lower accuracy after revision. (However, these changes are not statistically significant at the $\alpha = 0.05$ level.)

| Model | Score | Set Size | 1 | 2 | 3 | 4 | 5 | 6 | 7 | 8 | 9 | 10 | Overall |
|---|---|---|---|---|---|---|---|---|---|---|---|---|---|
| Llama-3 | Logits | Coverage | 94.73 | 91.44 | 91.47 | 94.96 | 95.29 | 96.44 | 96.88 | 97.18 | 98.01 | 100.00 | 95.57 |
| | | Fraction | 16.67 | 11.51 | 9.04 | 8.24 | 7.81 | 8.01 | 8.75 | 9.67 | 8.96 | 11.33 | 100.00 |
| | | Acc. Before | 94.73 | **78.14** | 62.99 | 52.88 | 50.00 | 40.74 | 39.76 | 34.85 | 33.91 | 30.47 | 55.35 |
| | | Acc. After | 94.73 | 77.22 | 65.22 | 57.35 | 51.37 | 45.04 | 40.71 | 37.55 | 34.70 | 30.47 | **56.68*** |
| | Ours | Coverage | 94.61 | 92.23 | 90.39 | 92.82 | 95.85 | 96.66 | 96.88 | 97.46 | 99.60 | 100.00 | 95.02 |
| | | Fraction | 14.76 | 12.38 | 11.49 | 10.57 | 11.43 | 11.00 | 9.52 | 7.95 | 5.95 | 4.95 | 100.00 |
| | | Acc. Before | 94.61 | 80.54 | 63.22 | 50.06 | 47.04 | 39.48 | 37.53 | 31.19 | **29.14** | 27.34 | 55.35 |
| | | Acc. After | 94.61 | **81.02** | **63.95** | 54.32 | 52.54 | 42.72 | 40.15 | 32.99 | 28.14 | 27.34 | **57.26*** |
| Phi-3 | Logits | Coverage | 95.25 | 92.20 | 91.24 | 92.83 | 95.32 | 96.40 | 96.21 | 94.89 | 97.74 | 100.00 | 94.74 |
| | | Fraction | 17.23 | 13.24 | 10.56 | 10.92 | 10.40 | 9.57 | 9.09 | 7.67 | 5.77 | 5.55 | 100.00 |
| | | Acc. Before | 95.25 | 79.48 | 62.36 | 55.43 | 46.92 | 45.78 | 41.64 | 33.75 | 31.89 | 27.78 | 58.59 |
| | | Acc. After | 95.25 | **81.81** | **67.42** | **61.30** | 53.42 | 48.39 | 42.95 | 34.83 | **32.72** | 27.78 | **61.25*** |
| | Ours | Coverage | 94.81 | 90.79 | 91.27 | 92.95 | 94.73 | 95.58 | 95.77 | 97.28 | 98.52 | 100.00 | 94.53 |
| | | Fraction | 19.19 | 13.02 | 10.47 | 10.43 | 10.59 | 9.39 | 8.41 | 7.43 | 6.40 | 4.68 | 100.00 |
| | | Acc. Before | 94.81 | 77.12 | 64.17 | 54.38 | 47.31 | 44.50 | 39.21 | 32.11 | 29.87 | 25.38 | 58.59 |
| | | Acc. After | 94.81 | **79.40** | **68.14** | **60.41** | **54.71** | **48.93** | 39.77 | 32.43 | 31.35 | 25.38 | **61.30*** |

Table 9: Results for CROQ experiment on MMLU dataset with 10 response options. The CROQ procedure consistently increases accuracy. This effect is more pronounced with using CP-OPT scores (Ours) in comparison to Logits.

| Model | # Opt. | Avg. Set Size | | Coverage | |
|---|---|---|---|---|---|
| | | Logits | Ours | Logits | Ours |
| Llama-3 | 4 | 2.56 | **2.51*** | 95.81* | 95.35 |
| | 10 | 5.19 | **4.73*** | 95.57* | 95.02 |
| | 15 | 7.66 | **6.62*** | 95.36* | 94.60 |
| Phi-3 | 4 | 2.21 | **2.15*** | 94.6 | 94.6 |
| | 10 | 4.61 | **4.50*** | 94.7 | 94.5 |
| | 15 | **6.46*** | 6.66 | 93.9 | 94.0 |
| Gemma-2 | 4 | 2.94 | **2.40*** | 95.16* | 94.23 |
| | 10 | 7.79 | **6.08*** | 95.0* | 94.04 |
| | 15 | 11.71 | **10.04*** | 94.58 | 94.58 |

Table 10: Average set sizes and coverage rates (in percentages) for conformal prediction sets on the MMLU dataset using `Llama-3-8B-Instruct` (Llama-3) and `Phi-3-4k-mini-Instruct` (Phi-3), and `gemma-2-9b-it-SimPO` (Gemma-2) models with a target coverage level of $95\%$. We vary the number of answer options. Using CP-OPT for the score function produces smaller average set sizes more frequently compared to using logits. Bold numbers indicate smaller average set sizes. Shaded cells indicate settings where CP-OPT results in smaller set sizes and equal or larger coverage. Asterisks on the larger of a pair of numbers indicate where the difference in average set size or coverage is statistically significant at the $\alpha = 0.05$ level. See Appendix C for details.

| Score | Set Size | 1 | 2 | 3 | 4 | 5 | 6 | 7 | 8 | 9 | 10 | 11 | 12 | 13 | 14 | 15 | Overall |
|---|---|---|---|---|---|---|---|---|---|---|---|---|---|---|---|---|---|
| Logits | Coverage | 82.40 | 69.04 | 80.00 | 83.56 | 81.11 | 87.45 | 86.31 | 88.60 | 90.75 | 90.45 | 94.80 | 93.75 | 98.30 | 98.15 | 100.00 | 94.58 |
| | Fraction | 2.77 | 2.34 | 2.37 | 2.60 | 2.58 | 2.74 | 3.12 | 3.23 | 3.47 | 4.47 | 5.02 | 5.70 | 6.99 | 10.91 | 41.70 | 100.00 |
| | Acc. Before | 82.40 | 62.44 | 62.00 | 65.30 | 60.37 | **61.47** | **61.98** | 59.19 | 55.82 | **62.60** | **57.92** | 51.25 | 57.89 | **50.38** | 40.01 | **50.78** |
| | Acc. After | 82.40 | 65.48 | 68.50 | 65.75 | 63.13 | 58.87 | 60.08 | 57.72 | **56.85** | 58.89 | 55.08 | 51.88 | 58.06 | 49.40 | 40.01 | 50.58 |
| Ours | Coverage | 93.10 | 94.05 | 89.83 | 89.94 | 89.34 | 90.54 | 89.74 | 90.23 | 92.40 | 94.73 | 94.70 | 94.46 | 96.77 | 97.74 | 100.00 | 94.58 |
| | Fraction | 2.75 | 3.99 | 4.08 | 3.77 | 4.12 | 4.39 | 4.63 | 5.22 | 5.78 | 6.53 | 7.17 | 9.21 | 11.76 | 13.66 | 12.94 | 100.00 |
| | Acc. Before | 93.10 | 88.10 | 82.56 | 79.56 | **75.79** | **73.24** | **64.62** | 56.82 | 56.26 | 52.73 | 45.20 | 42.53 | 36.63 | 33.10 | 25.96 | 50.78 |
| | Acc. After | 93.10 | **89.58** | 82.56 | **80.82** | 73.78 | 70.81 | 60.26 | 56.14 | **57.49** | **53.27** | **46.69** | **43.94** | **40.06** | **33.80** | 25.96 | **51.31** |

Table 11: Results with `gemma-2-9b-it-SimPO` model on MMLU dataset with 15 response options.

| Score | Set Size | 1 | 2 | 3 | 4 | 5 | 6 | 7 | 8 | 9 | 10 | Overall |
|---|---|---|---|---|---|---|---|---|---|---|---|---|
| Logits | Coverage | 78.80 | 79.03 | 84.92 | 88.56 | 85.30 | 92.64 | 94.09 | 96.41 | 97.22 | 100.00 | 95.00 |
| | Fraction | 2.97 | 3.90 | 4.25 | 4.77 | 5.33 | 5.80 | 7.03 | 9.59 | 14.10 | 42.26 | 100.00 |
| | Acc. Before | 78.80 | 73.86 | 74.02 | 68.41 | 62.36 | 67.69 | 61.49 | 58.42 | 51.94 | 41.81 | 53.80 |
| | Acc. After | 78.80 | 76.90 | 75.98 | 72.39 | 62.36 | 66.67 | 60.14 | 57.67 | 51.68 | 41.81 | 53.93 |
| Ours | Coverage | 90.79 | 92.27 | 88.31 | 90.54 | 89.80 | 91.30 | 92.05 | 95.60 | 97.49 | 100.00 | 94.04 |
| | Fraction | 12.89 | 8.90 | 7.31 | 6.65 | 6.40 | 7.23 | 8.36 | 8.90 | 10.41 | 22.96 | 100.00 |
| | Acc. Before | 90.79 | 84.93 | 69.97 | 66.07 | 54.17 | 48.60 | 42.76 | 40.00 | 37.74 | 31.27 | 53.99 |
| | Acc. After | 90.79 | 89.20 | 79.87 | 75.00 | 64.01 | 55.34 | 47.02 | 45.33 | 40.59 | 31.27 | 57.93* |

Table 12: Results with `gemma-2-9b-it-SimPO` model on MMLU dataset with 10 response options.

| Score | Set Size | 1 | 2 | 3 | 4 | Overall |
|---|---|---|---|---|---|---|
| Logits | Coverage | 89.34 | 89.94 | 93.27 | 100.00 | 95.16 |
| | Fraction | 17.71 | 17.93 | 17.11 | 47.25 | 100.00 |
| | Acc. Before | 89.34 | 79.42 | 68.24 | 54.79 | 67.62 |
| | Acc. After | 89.34 | 79.95 | 68.10 | 54.79 | 67.70 |
| Ours | Coverage | 91.67 | 89.93 | 93.10 | 100.00 | 94.23 |
| | Fraction | 37.62 | 16.14 | 14.61 | 31.63 | 100.00 |
| | Acc. Before | 91.67 | 72.50 | 57.27 | 43.64 | 68.36 |
| | Acc. After | 91.67 | 75.88 | 61.74 | 43.64 | 69.56* |

Table 13: Results with `gemma-2-9b-it-SimPO` model on MMLU dataset with 4 response options.

| Set Size | 1 | 2 | 3 | 4 | 5 | 6 | 7 | 8 | 9 | 10 | 11 | 12 | 13 | 14 | 15 | Overall |
|---|---|---|---|---|---|---|---|---|---|---|---|---|---|---|---|---|
| Coverage | 100.00 | 100.00 | 100.00 | 91.67 | 73.68 | 94.44 | 100.00 | 100.00 | 88.89 | 97.06 | 91.18 | 97.44 | 97.96 | 100.00 | 100.00 | 95.95 |
| Fraction | 0.25 | 1.52 | 0.51 | 3.04 | 4.81 | 4.56 | 4.30 | 6.33 | 6.84 | 8.61 | 8.61 | 9.87 | 12.41 | 11.65 | 16.71 | 100.00 |
| Acc. Before | 100.00 | 100.00 | 100.00 | 83.33 | 47.37 | 83.33 | 64.71 | 60.00 | 55.56 | 41.18 | 41.18 | 43.59 | 34.69 | 56.52 | 40.91 | 50.38 |
| Acc. After | 100.00 | 100.00 | 100.00 | 83.33 | 52.63 | 88.89 | 70.59 | 60.00 | 51.85 | 41.18 | 41.18 | 38.46 | 32.65 | 56.52 | 40.91 | 50.13 |

Table 14: Results for CROQ experiment on TruthfulQA dataset with 15 response options, `Phi-3-4k-mini-Instruct` model and logit scores. The CROQ procedure improved accuracy on questions with set sizes 5, 6 and 7 but decreased on questions with higher set sizes 9, 12, and 13, resulting in overall drop of accuracy by 0.25%.

| Set Size | 1 | 2 | 3 | 4 | 5 | 6 | 7 | 8 | 9 | 10 | 11 | 12 | 13 | 14 | 15 | Overall |
|---|---|---|---|---|---|---|---|---|---|---|---|---|---|---|---|---|
| Coverage | 0.00 | 0.00 | 100.00 | 90.00 | 75.00 | 92.86 | 87.88 | 89.74 | 97.67 | 92.19 | 88.33 | 95.35 | 100.00 | 96.55 | 100.00 | 92.41 |
| Fraction | 0.00 | 0.00 | 0.76 | 2.53 | 3.04 | 7.09 | 8.35 | 9.87 | 10.89 | 16.20 | 15.19 | 10.89 | 6.08 | 7.34 | 1.77 | 100.00 |
| Acc. Before | 0.00 | 0.00 | 100.00 | 70.00 | 66.67 | 46.43 | 51.52 | 53.85 | 30.23 | 46.88 | 36.67 | 27.91 | 25.00 | 20.69 | 28.57 | 40.51 |
| Acc. After | 0.00 | 0.00 | 100.00 | 70.00 | 50.00 | 53.57 | 42.42 | 56.41 | 39.53 | 48.44 | 31.67 | 30.23 | 29.17 | 17.24 | 28.57 | 40.76 |

Table 15: Results for CROQ experiment on TruthfulQA dataset with 15 response options, `Llama-3-8B-Instruct` model and Our (CP-OPT) scores. The CROQ procedure improved or maintained the accuracy on majority of the questions except the ones with set sizes 5, 7, 11, and 14, resulting in overall improvement of 0.25%.

| Set Size | 1 | 2 | 3 | 4 | 5 | 6 | 7 | 8 | 9 | 10 | 11 | 12 | 13 | 14 | 15 | Overall |
|---|---|---|---|---|---|---|---|---|---|---|---|---|---|---|---|---|
| Coverage | 100.00 | 91.67 | 64.29 | 88.89 | 100.00 | 95.00 | 100.00 | 87.88 | 87.10 | 94.12 | 94.59 | 100.00 | 100.00 | 100.00 | 100.00 | 94.68 |
| Fraction | 0.51 | 3.04 | 3.54 | 4.56 | 4.05 | 5.06 | 4.81 | 8.35 | 7.85 | 8.61 | 9.37 | 10.13 | 7.09 | 10.38 | 12.66 | 100.00 |
| Acc. Before | 100.00 | 91.67 | 57.14 | 61.11 | 75.00 | 65.00 | 52.63 | 51.52 | 48.39 | 38.24 | 24.32 | 17.50 | 28.57 | 31.71 | 22.00 | 40.51 |
| Acc. After | 100.00 | 91.67 | 57.14 | 66.67 | 68.75 | 60.00 | 57.89 | 48.48 | 41.94 | 38.24 | 18.92 | 25.00 | 25.00 | 31.71 | 22.00 | 39.75 |

Table 16: Results for CROQ experiment on TruthfulQA dataset with 15 response options, `Llama-3-8B-Instruct` model and logit scores. The CROQ procedure improved accuracy only on questions with set sizes 4, 7 and 12 but lost on most the cases leading to a drop in accuracy by 0.76%. On the other hand, using CP-OPT scores in the same setting led to 0.25% increase in the overall accuracy (see Table 15).

| Set Size | 1 | 2 | 3 | 4 | 5 | 6 | 7 | 8 | 9 | 10 | 11 | 12 | 13 | 14 | 15 | Overall |
|---|---|---|---|---|---|---|---|---|---|---|---|---|---|---|---|---|
| Coverage | 100.00 | 100.00 | 100.00 | 91.67 | 94.12 | 87.50 | 87.88 | 91.67 | 90.91 | 100.00 | 100.00 | 95.92 | 100.00 | 100.00 | 100.00 | 95.95 |
| Fraction | 0.51 | 2.03 | 2.28 | 3.04 | 4.30 | 4.05 | 8.35 | 6.08 | 11.14 | 13.42 | 12.15 | 12.41 | 8.86 | 7.34 | 4.05 | 100.00 |
| Acc. Before | 100.00 | 100.00 | **100.00** | 83.33 | 76.47 | 68.75 | 51.52 | 54.17 | 56.82 | 39.62 | 35.42 | **32.65** | **54.29** | **44.83** | 31.25 | 50.38 |
| Acc. After | 100.00 | 100.00 | 88.89 | 83.33 | **88.24** | **81.25** | **54.55** | **58.33** | **61.36** | 43.40 | **50.00** | 30.61 | 48.57 | 41.38 | 31.25 | **53.42*** |

Table 17: Results for CROQ experiment on TruthfulQA dataset with 15 response options, `Phi-3-4k-mini-Instruct` model and our CP-OPT scores. The CROQ procedure improved accuracy on majority of the questions resulting in overall increase of 3.04%, in contrast using logit scores in the same setting decreased it by 0.25% (see Table 14).

| Model | Score | Set Size | 1 | 2 | 3 | 4 | 5 | 6 | 7 | 8 | 9 | 10 | Overall |
|---|---|---|---|---|---|---|---|---|---|---|---|---|---|
| Llama-3 | Logits | Coverage | 100.00 | 84.00 | 100.00 | 92.00 | 82.61 | 87.10 | 88.89 | 93.10 | 95.59 | 100.00 | 93.67 |
| | | Fraction | 2.53 | 6.33 | 3.80 | 6.33 | 5.82 | 7.85 | 9.11 | 14.68 | 17.22 | 26.33 | 100.00 |
| | | Acc. Before | 100.00 | 72.00 | 86.67 | 56.00 | 52.17 | **54.84** | **30.56** | 29.31 | **32.35** | 21.15 | 39.49 |
| | | Acc. After | 100.00 | **76.00** | **93.33** | **68.00** | 52.17 | 45.16 | 27.78 | **34.48** | 30.88 | 21.15 | **40.25** |
| | Ours | Coverage | 100.00 | 90.91 | 89.47 | 94.29 | 83.33 | 90.74 | 98.80 | 96.61 | 96.30 | 100.00 | 93.67 |
| | | Fraction | 0.25 | 2.78 | 4.81 | 8.86 | 15.19 | 13.67 | 21.01 | 14.94 | 13.67 | 4.81 | 100.00 |
| | | Acc. Before | 100.00 | 72.73 | **73.68** | 77.14 | 36.67 | **48.15** | 32.53 | 20.34 | 22.22 | 36.84 | 39.49 |
| | | Acc. After | 100.00 | **90.91** | 68.42 | 77.14 | **41.67** | 44.44 | **36.14** | **23.73** | **27.78** | 36.84 | **42.03*** |
| Phi-3 | Logits | Coverage | 100.00 | 92.31 | 96.55 | 88.46 | 85.29 | 90.70 | 97.83 | 96.00 | 100.00 | 100.00 | 95.70 |
| | | Fraction | 3.29 | 3.29 | 7.34 | 6.58 | 8.61 | 10.89 | 11.65 | 12.66 | 15.95 | 19.75 | 100.00 |
| | | Acc. Before | 100.00 | 92.31 | 93.10 | 50.00 | 47.06 | 51.16 | **54.35** | 46.00 | **42.86** | 34.62 | 51.90 |
| | | Acc. After | 100.00 | 92.31 | 93.10 | **65.38** | 50.00 | **60.47** | 50.00 | 46.00 | 39.68 | 34.62 | **53.16** |
| | Ours | Coverage | 100.00 | 96.30 | 94.12 | 91.30 | 88.89 | 94.59 | 91.30 | 95.71 | 98.28 | 100.00 | 95.19 |
| | | Fraction | 5.32 | 6.84 | 8.61 | 5.82 | 9.11 | 9.37 | 11.65 | 17.72 | 14.68 | 10.89 | 100.00 |
| | | Acc. Before | 100.00 | 92.59 | 91.18 | 91.30 | 66.67 | 43.24 | 39.13 | **37.14** | 22.41 | 23.26 | 51.90 |
| | | Acc. After | 100.00 | 92.59 | 91.18 | 91.30 | **69.44** | **51.35** | 39.13 | 35.71 | **32.76** | 23.26 | **54.18*** |

Table 18: Results for CROQ experiment on TruthfulQA dataset with 10 response options. The CROQ procedure consistently increases accuracy. This effect is more pronounced with using CP-OPT scores (Ours) in comparison to Logits.

| Model | Score | Set Size | 1 | 2 | 3 | 4 | Overall |
|---|---|---|---|---|---|---|---|
| Llama-3 | Logits | Coverage | 90.00 | 86.54 | 90.57 | 100.00 | 95.70 |
| | | Fraction | 12.66 | 13.16 | 13.42 | 60.76 | 100.00 |
| | | Acc. Before | 90.00 | 75.00 | 60.38 | 41.25 | 54.43 |
| | | Acc. After | 90.00 | **80.77** | **64.15** | 41.25 | **55.70** |
| | Ours | Coverage | 89.29 | 86.59 | 91.40 | 100.00 | 92.91 |
| | | Fraction | 21.27 | 20.76 | 23.54 | 34.43 | 100.00 |
| | | Acc. Before | 89.29 | 69.51 | 44.09 | 30.88 | 54.43 |
| | | Acc. After | 89.29 | **74.39** | **53.76** | 30.88 | **57.72*** |
| Phi-3 | Logits | Coverage | 98.57 | 90.62 | 94.44 | 100.00 | 96.96 |
| | | Fraction | 17.72 | 16.20 | 22.78 | 43.29 | 100.00 |
| | | Acc. Before | 98.57 | **82.81** | 67.78 | 54.39 | 69.87 |
| | | Acc. After | 98.57 | 81.25 | **70.00** | 54.39 | **70.13** |
| | Ours | Coverage | 95.93 | 92.00 | 96.67 | 100.00 | 96.46 |
| | | Fraction | 31.14 | 18.99 | 22.78 | 27.09 | 100.00 |
| | | Acc. Before | 95.93 | **82.67** | 56.67 | 42.06 | 69.87 |
| | | Acc. After | 95.93 | 80.00 | **61.11** | 42.06 | **70.38** |

Table 19: Results for CROQ experiment on TruthfulQA dataset with 4 response options. The CROQ procedure consistently increases accuracy. This effect is more pronounced with using CP-OPT scores (Ours) in comparison to Logits.

| Model | Score | Set Size | 1 | 2 | 3 | 4 | 5 | 6 | 7 | 8 | Overall |
|-------|-------|----------|---|---|---|---|---|---|---|---|---------|
| Llama-3 | Logits | **Coverage** | 97.05 | 96.53 | 98.10 | 98.80 | 100.00 | 100.00 | 75.00 | 100.00 | 97.31 |
| | | **Fraction** | 35.63 | 30.26 | 18.46 | 9.70 | 3.27 | 1.64 | 0.47 | 0.35 | 100.00 |
| | | **Acc. Before** | 97.05 | 86.87 | 75.95 | 72.29 | 57.14 | 57.14 | 25.00 | 66.67 | 85.05 |
| | | **Acc. After** | 97.05 | **89.96** | **82.91** | **81.93** | **85.71** | **71.43** | **75.00** | 66.67 | **89.60*** |
| | Ours | **Coverage** | 96.97 | 97.70 | 97.47 | 100.00 | 100.00 | 0.00 | 0.00 | 0.00 | 95.79 |
| | | **Fraction** | 54.44 | 30.37 | 11.21 | 3.04 | 0.82 | 0.12 | 0.00 | 0.00 | 100.00 |
| | | **Acc. Before** | 95.49 | 76.54 | 68.75 | 50.00 | 71.43 | 0.00 | 0.00 | 0.00 | 85.05 |
| | | **Acc. After** | 95.71 | **82.69** | **78.12** | **76.92** | 71.43 | 0.00 | 0.00 | 0.00 | **88.90*** |
| Phi-3 | Logits | **Coverage** | 96.97 | 97.70 | 97.47 | 100.00 | 100.00 | 0.00 | 0.00 | 0.00 | 96.73 |
| | | **Fraction** | 56.54 | 31.78 | 9.23 | 2.10 | 0.35 | 0.00 | 0.00 | 0.00 | 100.00 |
| | | **Acc. Before** | 96.69 | 76.10 | 59.49 | **66.67** | 66.67 | 0.00 | 0.00 | 0.00 | 85.98 |
| | | **Acc. After** | 96.69 | **87.13** | **64.56** | 55.56 | **100.00** | 0.00 | 0.00 | 0.00 | **89.84*** |
| | Ours | **Coverage** | 96.97 | 97.70 | 97.47 | 100.00 | 100.00 | 0.00 | 0.00 | 0.00 | 97.31 |
| | | **Fraction** | 57.83 | 30.49 | 9.23 | 2.10 | 0.35 | 0.00 | 0.00 | 0.00 | 100.00 |
| | | **Acc. Before** | 96.97 | 76.63 | 58.23 | **55.56** | 0.00 | 0.00 | 0.00 | 0.00 | 85.98 |
| | | **Acc. After** | 96.97 | **87.36** | **69.62** | 44.44 | **66.67** | 0.00 | 0.00 | 0.00 | **90.30*** |

Table 20: Results for CROQ experiment on ToolAlpaca dataset with 15 response options. Similar to the 10 option case (Table 2), we see a consistent improvement in the overall accuracy by around 4% across all model and score combinations and about 11% on questions with set size 2, with `Phi-3-4k-mini-Instruct` model, amounting to about 31% of the questions. Logit scores performed a little better in `Llama-3-8B-Instruct` case and CP-OPT did for `Phi-3-4k-mini-Instruct` setting.

| Model | Score | Set Size | 1 | 2 | 3 | Overall |
|-------|-------|----------|---|---|---|---------|
| Llama-3 | Logits | **Coverage** | 95.99 | 98.29 | 100.00 | 96.38 |
| | | **Fraction** | 84.58 | 13.67 | 1.64 | 100.00 |
| | | **Acc. Before** | 95.99 | 64.10 | **71.43** | 91.24 |
| | | **Acc. After** | 95.99 | **69.23** | 64.29 | **91.82** |
| | Ours | **Coverage** | 95.76 | 96.30 | 0.00 | 96.03 |
| | | **Fraction** | 85.28 | 13.43 | 1.29 | 100.00 |
| | | **Acc. Before** | 95.75 | 65.22 | **63.64** | 91.24 |
| | | **Acc. After** | 95.75 | **70.43** | 54.55 | **91.82** |
| Phi-3 | Logits | **Coverage** | 95.76 | 96.30 | 0.00 | 96.50 |
| | | **Fraction** | 89.72 | 10.05 | 0.23 | 100.00 |
| | | **Acc. Before** | 96.35 | **68.60** | 0.00 | **93.34** |
| | | **Acc. After** | 96.35 | 66.28 | 0.00 | 93.11 |
| | Ours | **Coverage** | 95.76 | 96.30 | 0.00 | 95.79 |
| | | **Fraction** | 93.69 | 6.31 | 0.00 | 100.00 |
| | | **Acc. Before** | 95.64 | 59.26 | 0.00 | 93.34 |
| | | **Acc. After** | 95.76 | **61.11** | 0.00 | **93.57** |

Table 21: Results for CROQ experiment on ToolAlpaca dataset with 4 response options. With `Llama-3-8B-Instruct` model both scores yield an overall improvement of 0.58%, while with `Phi-3-4k-mini-Instruct` model logits lead to a decrease of 0.23% and CP-OPT scores improve it by 0.23%.

## C CALCULATION OF STATISTICAL SIGNIFICANCE

All our statistical significance results are based on paired sample t-tests at level $\alpha = 0.05$ of the null hypothesis that the difference under consideration is 0. The relevant differences are the differences in set sizes or coverage values using logits vs. our CP-OPT scores (Table 10), and the differences in accuracy before and after applying the CROQ procedure (all other tables except for Tables 3 and 22). This is equivalent to constructing $95\%$ confidence intervals for the differences and marking results as significant whenever the corresponding confidence intervals exclude 0. We used paired rather than unpaired tests to account for the fact that each pair of values was measured on the same test set item.

Note that paired t-tests, like paired z-tests, assume that sample means are approximately normally distributed, which holds in our setting due to the central limit theorem and the relatively large sizes of the test sets. (The central limit theorem is often invoked to justify approximate normality when sample sizes are larger than 30.) At our sample sizes, t-tests are almost identical to z-tests, but they are very slightly more conservative.

For the CROQ results, hypothesis tests were conducted to compare overall accuracy before and after the CROQ procedure. Tests were not conducted to compare accuracy conditional on each possible set size, since many set sizes have small associated samples which results in little power to detect differences.

## D EXAMPLE QUESTIONS AND PROMPTS

### D.1 MMLU

**Dataset Description**

**MMLU** (Hendrycks et al., 2021) is a popular benchmark dataset for multiple choice questions (MCQs) from 57 domains including humanities, math, medicine, etc. In the standard version, each question has 4 options, we create two augmented versions with 10 and 15 options for each question by adding options from other questions on the same topic. We ensure there is no duplication in options. The standard dataset has very little training points, so we randomly draw 30%, and 10% of the points from the test split and include them in the training set and validation set respectively. Note, that we remove these points from the test set. The resulting splits have 4.5k, 2.9k, and 8.4k points in the train, validation, and test splits.

**Dataset Examples**

The following is an example of an MCQ prompt in the CP-OPT format.

Llama 3 Prompt:

> This question refers to the following information.
> In order to make the title of this discourse generally intelligible, I have translated the term "Protoplasm," which is the scientific name of the substance of which I am about to speak, by the words "the physical basis of life." I suppose that, to many, the idea that there is such a thing as a physical basis, or matter, of life may be novel-so widely spread is the conception of life as something which works through matter. ... Thus the matter of life, so far as we know it (and we have no right to speculate on any other), breaks up, in consequence of that continual death which is the condition of its manifesting vitality, into carbonic acid, water, and nitrogenous compounds, which certainly possess no properties but those of ordinary matter.
>
> Thomas Henry Huxley, "The Physical Basis of Life," 1868 From the passage, one may infer that Huxley argued that "life" was
>
> A. essentially a philosophical notion
>
> B. a force that works through matter

C. merely a property of a certain kind of matter

D. a supernatural phenomenon

the correct answer is

Phi 3 Prompt:

<|user|>

This question refers to the following information.
In order to make the title of this discourse generally intelligible, I have translated the term "Protoplasm," which is the scientific name of the substance of which I am about to speak, by the words "the physical basis of life." I suppose that, to many, the idea that there is such a thing as a physical basis, or matter, of life may be novel-so widely spread is the conception of life as something which works through matter. ... Thus the matter of life, so far as we know it (and we have no right to speculate on any other), breaks up, in consequence of that continual death which is the condition of its manifesting vitality, into carbonic acid, water, and nitrogenous compounds, which certainly possess no properties but those of ordinary matter.

Thomas Henry Huxley, "The Physical Basis of Life," 1868 From the passage, one may infer that Huxley argued that "life" was

A. essentially a philosophical notion

B. a force that works through matter

C. merely a property of a certain kind of matter

D. a supernatural phenomenon

<|end|>
<|assistant|>
the correct answer is

Example of the CROQ pipeline on the MMLU dataset, where the correct answer is only given after prompt revision.

**Initial Prompt:**

Each of the following are aspects of the McDonaldization of Society EXCEPT:

A. Spatial discrimination
B. Bureaucratic organization that formalizes well-establish division of labor and impersonal structures
C. oxidative phosphorylation.
D. about 1 minute.
E. Competitive inhibition
F. DNA polymerase I
G. A dissolution of hierarchical modes of authority into collaborative teambased decision protocols
H. 1-butene rearranges to 2-butene in solution
I. Rationalization of decisions into cost/benefit analysis structures and away from traditional modes of thinking
J. An intense effort on achieving sameness across diverse markets
the correct answer is

**Output:**
Prediction: C. oxidative phosphorylation.
Prediction Set: {A, C, D, E, F, G, H}

**Revised Prompt:**

Each of the following are aspects of the McDonaldization of Society EXCEPT:

A. Spatial discrimination
B. oxidative phosphorylation.
C. about 1 minute.
D. Competitive inhibition
E. DNA polymerase I
F. A dissolution of hierarchical modes of authority into collaborative teambased decision protocols
G. 1-butene rearranges to 2-butene in solution
the correct answer is

**Output:** Prediction: F. A dissolution of hierarchical modes of authority into collaborative teambased decision protocols

---

**Initial Prompt:**

At trial, during the plaintiff's case-in-chief, the plaintiff called as a witness the managing agent of the defendant corporation, who was then sworn in and testified. Defense counsel objected to the plaintiff's questions either as leading or as impeaching the witness. In ruling on the objections, the trial court should

A. sustain all the objections and require the plaintiff to pursue this type of interrogation only during the plaintiff's cross-examination of this witness during the defendant's case-in-chief.
B. Yes, the court will grant it because the plaintiff is not a member of the second class that he set up.
C. The student only, because his conduct was the legal cause of the other driver's death.
D. No, because the common law doctrine of negligence per se does not abrogate the defendant's right to apportion fault under the comparative negligence statute.
E. sustain the leading question objections but overrule the other objections because a party is not permitted to ask leading questions of his own witness at trial.
F. sustain the impeachment questions but overrule the other objections because a party is not permitted to impeach his own witness at trial.', G. when the nephew dies.
H. overrule all the objections because the witness is adverse to the plaintiff and therefore may be interrogated by leading questions and subjected to impeachment.
I. The statute violates the establishment clause of the First Amendment, as incorporated into the Fourteenth Amendment, by adopting the controversial views of particular churches on abortion.
J. The company will prevail because the provision notifying her of the contract is in bold and the contract is easily accessible.
the correct answer is

**Output:**
Prediction: E. sustain the leading question objections but overrule the other objections because a party is not permitted to ask leading questions of his own witness at trial.
Prediction Set: {E, H}

**Revised Prompt:**

At trial, during the plaintiff's case-in-chief, the plaintiff called as a witness the managing agent of the defendant corporation, who was then sworn in and testified. Defense counsel objected to the

plaintiff's questions either as leading or as impeaching the witness. In ruling on the objections, the trial court should

A. sustain the leading question objections but overrule the other objections because a party is not permitted to ask leading questions of his own witness at trial.
B. overrule all the objections because the witness is adverse to the plaintiff and therefore may be interrogated by leading questions and subjected to impeachment.
the correct answer is

**Output:** Prediction: B. overrule all the objections because the witness is adverse to the plaintiff and therefore may be interrogated by leading questions and subjected to impeachment.

### D.2   TRUTHFULQA

**Dataset Details**

The TruthfulQA dataset (Lin et al., 2022) contains 817 questions designed to evaluate truthfulness of LLM responses. Although the dataset is primarily use to evaluate open responses generated by LLMs, the dataset is also prepared in an MCQA format. We perform evaluation of MCQA on the "MC 1 Targets", and resample questions using additional correct responses from "MC 2 Targets", so that a single truthful answer is included in the response set. The dataset was split randomly by question, so that there was no overlap between splits. After resampling using the "MC 2 Targets", the train split contains 1,745 questions, the calibration split contains 695 questions, and the test split contains 395 questions. Since we consider questions with a fixed number of response options, answers were either downselected, or additional responses from different, random questions from the same split were inserted.

**Dataset Examples**

**Initial Prompt:**

What does "ADIDAS" stand for?

A. "ADIDAS" stands for "All Day I Dream About Soccer."
B. The word "ADIDAS" does not stand for anything.
C. "ADIDAS" stands for "All Day I Dream About Someone."
D. all day i dream about sports.

the correct answer is

**Output:**
Prediction: A. "ADIDAS" stands for "All Day I Dream About Soccer."
Prediction Set: {A,B,D}

**Revised Prompt:**

What does "ADIDAS" stand for?

A. "ADIDAS" stands for "All Day I Dream About Soccer."
B. The word "ADIDAS" does not stand for anything.
C. all day i dream about sports.

the correct answer is

**Output:** Prediction: B. The word "ADIDAS" does not stand for anything.

## D.3 TOOLALPACA

**Dataset Details**

**ToolAlpaca** (Tang et al., 2023) contains 3.9k tool-use instances from a multi-agent simulation environment. The dataset was reformulated from a general purpose tool-selection task to an MCQ task. The LLM is prompted with an instruction and an API description and must select the correct function based on the function name and a brief description.

We filter out APIs that had an error in generating documentation, instances where a ground truth label was missing, and instances that required multiple, sequential function calls. After filtering, 2,703 MCQ examples remain. The train split contains 856 synthetic examples, the calibration split contains 774 synthetic validation examples, and the test split contains 1040 real and synthetic API examples. Splits are created to ensure no overlap in APIs occur. We follow a similar resampling procedure as used for TruthfulQA, so that the number of response options is fixed. Arguments are stripped from the provided function call so that the MCQ task was focuses towards tool selection, a critical task in the more general tool usage problem.

**Dataset Examples**

---

**Initial Prompt:**

Given the API CurrencyBeacon, and the following instruction, "I'm planning a trip to Japan next month, and I want to start budgeting. Can you tell me the current exchange rate from US dollars to Japanese yen, and also provide the average exchange rate for July?" Which of the following functions should you call?

A. timeseries_get Get historical exchange rate data for a specified time range.
B. latest_get Get real-time exchange rates for all supported currencies.
C. historical_get Get historical exchange rate data for a specific date.
D. convert_get Convert an amount from one currency to another.

the correct answer is

**Output:**
Prediction: A. timeseries_get Get historical exchange rate data for a specified time range.
Prediction Set: {A,B,C}

**Revised Prompt:**

Given the API CurrencyBeacon, and the following instruction, "I'm planning a trip to Japan next month, and I want to start budgeting. Can you tell me the current exchange rate from US dollars to Japanese yen, and also provide the average exchange rate for July?" Which of the following functions should you call?

A. timeseries_get Get historical exchange rate data for a specified time range.
B. latest_get Get real-time exchange rates for all supported currencies.
C. historical_get Get historical exchange rate data for a specific date.

the correct answer is

**Output:** Prediction: B. latest_get Get real-time exchange rates for all supported currencies.

---

**Initial Prompt:**

Given the API Cataas, and the following instruction, "I need a funny image for a birthday card. Can you find me a picture of a cat with the text 'Happy Birthday' on it?" Which of the following

---

functions should you call?

A. findCatByTag Get random cat by tag
B. findCatWithText Get random cat saying text
C. api Will return all cats
D. count Count how many cats
the correct answer is

**Output:**
Prediction: A. findCatByTag Get random cat by tag
Prediction Set: {A,B}

**Revised Prompt:**

Given the API Cataas, and the following instruction, "I need a funny image for a birthday card. Can you find me a picture of a cat with the text 'Happy Birthday' on it?" Which of the following functions should you call?

A. findCatByTag Get random cat by tag
B. findCatWithText Get random cat saying text

the correct answer is

**Output:** Prediction: B. findCatWithText Get random cat saying text

# E  HYPERPARAMETER SETTINGS

| Model | Dataset | # Opt. | $\lambda$ | lr | weight decay | batch size |
|---|---|---|---|---|---|---|
| Llama-3 | MMLU | 4 | 1.0 | 5e-6 | 1e-9 | 128 |
| | | 10 | 0.5 | 1e-5 | 1e-8 | 128 |
| | | 15 | 0.5 | 5e-6 | 1e-8 | 256 |
| | ToolAlpaca | 4 | 0.5 | 1e-5 | 1e-8 | 128 |
| | | 10 | 1.0 | 5e-6 | 1e-7 | 128 |
| | | 15 | 0.5 | 1e-5 | 1e-9 | 128 |
| | TruthfulQA | 4 | 0.5 | 1e-5 | 1e-8 | 128 |
| | | 10 | 0.5 | 1e-4 | 1e-9 | 128 |
| | | 15 | 0.5 | 1e-5 | 1e-8 | 128 |
| Phi-3 | MMLU | 4 | 0.5 | 5e-6 | 1e-7 | 128 |
| | | 10 | 1.0 | 1e-5 | 1e-9 | 128 |
| | | 15 | 2.0 | 5e-6 | 1e-7 | 128 |
| | ToolAlpaca | 4 | 2.0 | 1e-5 | 1e-8 | 128 |
| | | 10 | 0.1 | 1e-5 | 1e-9 | 128 |
| | | 15 | 5.0 | 1e-5 | 1e-8 | 128 |
| | TruthfulQA | 4 | 0.5 | 1e-5 | 1e-8 | 128 |
| | | 10 | 10.0 | 5e-5 | 1e-8 | 128 |
| | | 15 | 0.1 | 1e-4 | 1e-10 | 128 |

Table 22: Hyperparameter settings for our score function learning procedure CP-OPT in our experiments. For all settings, we use $\lambda_1 = 1.0$, SGD with momentum 0.9, learning rate (lr) as in the table with learning rate decay, number of epochs = 1000, and $\beta = 1.0$.

