# OpenReview forum: "Monty Hall and Optimized Conformal Prediction to Improve Decision-Making with LLMs"
_ICLR.cc/2025/Conference — Submitted to ICLR 2025_

### Official Review · Reviewer_pywK · 2024-10-21

**Soundness:** 2
**Presentation:** 2
**Contribution:** 2
**Rating:** 3
**Confidence:** 4

**Summary:**

The authors present two orthogonal methods related to conformal prediction: The first part of the manuscript introduces CP-OPT, an optimization framework to learn scores
that minimize set sizes. The second part introduces *conformal revision of questions
(CROQ)*, a method that improves model predictions by removing low-quality answers (the ones that are not part of the conformal prediction set) and querying the model again on the remaining high-quality examples. The methods are empirically assessed on three multiple-choice question answering tasks.

**Strengths:**

1. The ideas of the work are very interesting. In particular I find the second part of the manuscript about conformal revision of questions (CROQ) interesting.

2. The proposed methods are simple, orthogonal to the choice of underlying model architecture and easy to understand. The manuscript is accessible to readers with diverse backgrounds. I appreciate that the authors withstand the trend of overcomplicating their ideas.

3. I find it surprising that CROQ works. It shows that large language models are able to bootstrap in the sense that they can incrementally guide themselves to the correct answer, similar to the idea called *chain-of-thought reasoning* [1].

[1] Wei, Jason, et al. *Chain-of-thought prompting elicits reasoning in large language models.*. Advances in neural information processing systems 35 (2022): 24824-24837.

**Weaknesses:**

1. The manuscript is divided into two rather unrelated parts. Unfortunately, this division comes at the cost that both ideas are treated somewhat superficially. I would advise the authors to go for either one of two options: 1) Create a coherent and convincing story that unifies both parts or 2) Remove one of the two parts from the manuscript and extend the other part (I recommend option 2).

2. The writing of the manuscript can be improved. There exist a number of inconsistencies (see Questions section) and unclear/confusing notation (the latter point is likely subjective).

3. I am not convinced that CROQ works due to conformal prediction, as the $1 - \alpha$ coverage guarantee does not seem relevant for the method to work. For a downstream purpose, conformal prediction just yields an arbitrary threshold to remove low-quality answers. But it does not matter whether this prediction set achieves $1 - \alpha$ coverage, because this $\alpha$ has no particular meaning. Instead, the essential (and interesting) point seems to be that the model can improve its own predictions simply by removing answers that have low quality, where the quality threshold must be tuned. I hypothesize that it would be more efficient to tune this quality threshold directly rather than tuning the $\alpha$ parameter (which is subject to variance from the calibration set), which indirectly predicts a quality threshold. I would ask the authors to provide a convincing answer why the coverage guarantee at a given level $1 - \alpha$ is relevant for the method and why they cannot just tune the quality threshold directly.

**Questions:**

line 1: I am not sure whether the association with the Monty Hall Problem for CROQ is reasonable. There exists an essential difference: For CROQ, the agent (LLM) solves the answer entirely based on its own predictions (and data), whereas there exists an external entity in the Monty Hall Problem (the host) who opens a door. I suggest modifying the narrative.

line 38: *complete a task. (Qu et al., 2024;* The dot after *task* should be removed.

line 42: Figure 1 shows a large language model that makes a wrong prediction, but reveals little about what to expect when reading the manuscript. I would advise combining Figure 1 and Figure 2 or replacing Figure 1 by Figure 2 and omitting Figure 1 entirely (I recommend the latter).

line 46: In spite of the title, the work by [1] does not use conformal prediction, but a multiple hypothesis testing method called *learn then test* [2]. You may consider moving this reference somewhere else, removing the reference, or rewriting the sentence to broaden the scope.

line 53: *any scoring function* For the sake of consistency, I advise the authors to call this function *score function* everywhere.

line 175: Is *conformity score function* the same as *score function*, which is the same as *scoring function*? If yes, please replace all variants by *score function*.

line 178: I find the notation $C(\tilde{x} \, | \, g, \hat{\tau})$ a bit misleading. It may create the impression that the score function $g$ is a random variable. I would advise replacing this notation with something like $C(\tilde{x} ; g, \hat{\tau})$ or $C_{g, \hat{\tau}}(\tilde{x})$.

line 206: This notation seems quite unusual. I recommend just writing $ \mathbb{E}_{x} $ or e.g. $ \mathbb{E}_{x \sim P} $ (in the latter case,  $P$ must then also be introduced).

line 214: I am surprised to find that the optimal threshold $\tau$ is part of the definition. I believe that $\tau^*$ is just a deterministic function of $g^*$.

line 221: Replace the period with a colon: "*distribution.*" should be "*distribution:*".

line 229: Replace "*higher*" with "*larger*": "*higher $\beta$*" should be "*larger $\beta$*".

line 237: Specify what the arrow $\rightarrow$ means, e.g., *convergence in probability*.

line 240: What is this *flexible space of functions* $\mathcal{G}$?

line 245: Similarly as to my comment for line 214, I believe we only need to optimize for $g$, because $\tau$ is just a function of $g$.

line 245: The equation has $\lambda$ and $\lambda_2$. I suggest renaming $\lambda$ to $\lambda_1$ for sake of consistency.

line 245: Clarify what $|| g ||_2$ means. Is it the squared norm over parameters of $g$? Or is it $\int |g(x, y)| dxdy$?

line 245: Why is the penalty formulation used to solve a constrained optimization problem? Would it be possible to solve the problem using the augmented Lagrangian method, which leads to a more accurate solution?

line 289: Unlike the authors, I am surprised that this works: In contrast to the Monty Hall Problem, the door with the goat is not opened by an external entity, but by the model itself (which makes a great difference).

line 420: I believe that the *discussion* section should rather be called *results*.

line 479: [1] is a suitable reference in the more general context of risk control, but as mentioned in an earlier comment, [1] does not employ conformal prediction. I would therefore suggest removing the reference or rewriting the text accordingly.

line 484: As far as [1] is concerned, there seems to be another misunderstanding: [1] also aims at reducing prediction set size, unlike claimed. I would suggest removing this sentence.


[1] Quach, Victor, et al. *Conformal language modeling*. International Conference on Learning Representations, 2023.

[2] Angelopoulos, Anastasios N., et al. *Learn then test: Calibrating predictive algorithms to achieve risk control.* arXiv preprint arXiv:2110.01052, 2021.

---

> ### Author Response · Authors · 2024-11-23
> **Response to Reviewer pywK (Part 1)**
>
> We appreciate your careful reading and thorough feedback. Thank you for highlighting the strengths of our work, particularly the simplicity of our methods and the effectiveness of CROQ in improving LLM predictions. We’re especially pleased that, like us, you found the idea behind CROQ very interesting and appreciated its surprising results. We appreciate your suggestion connecting CROQ to chain-of-thought reasoning, which provides an exciting perspective. Below, we address the specific concerns raised.
>
>
> ### **1. Coherence of the methodologies**
>
> We understand the reviewer’s concern regarding the perceived separation between CP-OPT and CROQ. While these methods can function independently, they are complementary and align with our broader goal of robust uncertainty quantification and accuracy improvement in LLMs. To make this connection clearer, we have added Hypothesis H3 and additional experiments (Table 11, 12, 13) demonstrating how CP-OPT enhances CROQ by producing smaller, high-coverage prediction sets that improve LLM performance in the second round of querying.
>
>
> CROQ’s success depends on the size of the prediction sets it refines—smaller sets from CP-OPT reduce uncertainty and improve accuracy more effectively than larger sets from logits. Figure 4 further supports this by demonstrating that as prediction set size decreases (simulated using ground truth), CROQ's accuracy consistently improves. This evidence highlights the importance of CP-OPT in optimizing CROQ's performance. We believe this workflow provides a coherent and principled approach to reducing uncertainty and refining LLM predictions. We have revised the manuscript to highlight this connection more explicitly.
>
> ### **2.  Writing inconsistencies**
>
> We appreciate your careful reading and suggestions with line numbers. We have updated the paper to incorporate most of them. The updates are highlighted in blue color. We clarified what we mean by flexible $\mathcal{G}$ and provide details in the Appendix B.1. We updated the draft to consistently use ''score function'', use $C(x ; g,\tau)$  in place of $C(x | g,\tau)$. We also found Figure 1 was not adding much, so removed it.
>
> ### **3. On the value of conformal prediction in CROQ**
>
> We agree that if the sole goal is to optimize accuracy, one could directly tune a quality threshold as suggested by the reviewer. However, the suggested procedure is conceptually equivalent to what CP provides, with CP adding meaning and theoretical gurantees to the process. Moreover, CP serves our broader goal of uncertainty quantification and making LLM's inference robust.
>
> In our setup, LLMs produce an initial prediction set, which may then be refined and re-evaluated using the CROQ procedure. CP ensures that these sets are not only appropriately sized but also include the true answer with a specified probability (e.g., 95%). This ensures that CROQ operates on a solid theoretical foundation, balancing uncertainty reduction and accuracy improvement.
>
> Directly optimizing a threshold might yield similar results in some cases, but CP formalizes the process and guarantees coverage, making it a more versatile and reliable approach. By leveraging CP, we align CROQ with a well-established methodology, enhancing its interpretability and applicability to a wider range of tasks. We hope this clarifies the importance of CP in the context of CROQ.
>
>
> ### **4. Clarification on Monty Hall connection**
>
> While the Monty Hall analogy is not critical to the methodology, we use it to: (a) provide a familiar conceptual framework to understand the effectiveness of CROQ, and (b) highlight the broader possibilities for defining oracles in CROQ.
>
> In CROQ, the conformal set generated during the first stage contains the correct answer with a user-specified probability (e.g., 95%). This is conceptually similar to an oracle eliminating incorrect options ("goats") with high probability. The LLM is then re-queried with the remaining options, sometimes leading to improved accuracy by refining its predictions.
>
> Unlike Monty Hall, where the host provides definitive external knowledge, the "oracle" in CROQ is probabilistic and derives its knowledge from the conformal scores. These scores can be sourced from the same LLM or external models, making the process flexible.  We have updated the paper to clarify the analogy.
>
> [ Response continues in the next comment ]

---

> ### Author Response · Authors · 2024-11-23
> **Response to Reviewer pywK (Part 2)**
>
> ### **4. On Quach. et al. 2023**
>
> We appreciate the reviewer’s pointing out that there is a distinction between traditional conformal prediction and the learn-then-test framework, although we consider them to be closely related. (The work by Quach et al. (2023) describes their method as “Translating this process to conformal prediction” and “an extension of conformal prediction,” although they also draw contrasts between their method and conformal prediction.) For clarity, in the two places where we cite Quach et al., we will broaden the scope as suggested, replacing “conformal prediction” with “conformal prediction and related methods".
>
> ### **5. Optimization method for CP-OPT**
>
> While we introduce the penalty term $\lambda$ to transform the problem into an unconstrained one (P2), our approach treats $\lambda$ as a hyperparameter, unlike penalty or augmented Lagrangian methods, where this parameter is iteratively updated during optimization. We appreciate the suggestion to explore these methods for solving (P2) and recognize it as an interesting direction for future work.
>
> ### **6. Optimization over $g$ and $\tau$**
> You are correct that $\tau$ can be deterministically obtained once $g$ is specified. We include $\tau$ explicitly in the formulation to simplify the objective and avoid the complexity introduced by computing $\tau$ as the $\alpha$ quantile of $g$ during optimization. Estimating $\tau$ after each update of $g$ would significantly slow down the learning procedure for $g$.

---

> > ### Comment · Reviewer_pywK · 2024-11-24
> >
> > I thank the authors for their comprehensive response. I will comment on two points:
> >
> > Regarding 1. I agree that the two methods are complementary and can be used together. In my opinion, this is not a valid reason to put them into the same paper. I still believe that both ideas are treated too superficially. I also decided to take a closer look into the existing literature, motivated by the comment about lack of novelty brought up by reviewer *yss3*. It seems that CP-OPT is indeed very similar to [1] and [2].
> >
> > Regarding 3. This seems to be the great issue about QROC. I am afraid that the authors' statement *... CP adding meaning and theoretical gurantees to the process.* is incorrect: No guarantees are passed on from the conformal prediction set to the output of the model when you query it again, using the conformal prediction set (this is easy to see). Thus, QROC is an unnecessary detour for filtering out low-quality examples, which could be done in more straightforward ways. Consequently, I am afraid that the rationale of this method is flawed.
> >
> > In light of the two points above, I believe the manuscript would benefit from great revisions before it meets the publication criteria and I will decrease my score. For a future version of the manuscript, I would like to make two suggestions to the authors:
> >
> > 1. Leave out CP-OPT entirely or find a novel twist to it and then write a paper about that only.
> >
> > 2. Rewrite QROC as a method for chain of thought prompting, but leave out the conformal prediction part. Also, please make sure to properly check the related literature to ensure that your method is novel.
> >
> > I hope that this advice is helpful.
> >
> > [1] Cherian, John J., Isaac Gibbs, and Emmanuel J. Candès. "Large language model validity via enhanced conformal prediction methods." Advances in Neural Information Processing Systems (2024).
> >
> > [2] Stutz, David, Ali Taylan Cemgil, and Arnaud Doucet. "Learning optimal conformal classifiers." International Conference on Learning Representations (2022).

---

> > > ### Author Response · Authors · 2024-12-02
> > > **Response to Reviewer pywK**
> > >
> > > Here we respond to your three points,
> > >
> > > **1. Novelty**
> > >
> > > Firstly, we did discuss [1,2] in our paper how CP-OPT is related to them **(please see lines 174-179 and lines 491-495)**. We provide further clarification here,
> > >
> > > a. Stutz et al. 2022 [1] optimize similar objectives to CP-OPT but **during training of a classification model**. In contrast, our work aims to improve **post-hoc uncertainty quantification of LLMs for MCQ and tool-selection tasks**. Prior works using CP on MCQ tasks with LLMs have used heuristic or logit scores from LLMs. CP-OPT provides a principled solution to learn scores for these tasks and we provide extensive empirical evaluation of CP-OPT on MCQ and tool-selection tasks showing its effectiveness.
> > >
> > > b. Cherian et al. (2024) [2] focus on **factuality guarantees in open-ended generation tasks, where correctness is defined differently**. Their work redefines coverage around factuality or acceptability rather than correctness, as there may not be a single “correct” response. In contrast, CP-OPT is designed to generate the smallest possible set of response options while ensuring high probability inclusion of the correct answer, making it uniquely suited for finite-response tasks like MCQs. These differences were discussed in the paper (lines 491-495).
> > >
> > > To the best of our knowledge, these are novel contributions towards improving UQ and the accuracy of LLMs in finite response settings such as MCQ and tool-selection tasks.
> > >
> > >
> > > **2. Clarification on using CP in CROQ**
> > >
> > > We illustrate our reasoning to use CP in CROQ with the help of the following example. Consider a random black-box predictor, when given $M$ choices it selects one option randomly and outputs it as the answer, i.e. its probability of correctness is $1/M$. Now consider the following two scenarios,
> > >
> > > a. If there is a **deterministic oracle** that can reduce the choices to $m<M$, while ensuring that the true answer is among the $m$ choices, then the probability of correctness of the same random predictor would be $1/m$. Implying accuracy improvement $\Delta(M,m) = \frac{1}{m} - \frac{1}{M} = \frac{M-m}{mM} > 0$. The smaller the $m$ the larger the improvement.
> > >
> > > b. In practice, we do not have such a deterministic oracle that can reduce the choices to $m$ while retaining the true answer. Suppose, instead we have a  **"probababilistic oracle"** $\mathcal{P}$ that reduces the initial set of $M$ choices (for a randomly drawn question $x$) to $m_x$ while ensuring that the true answer is in the selected $m_x$ choices with probability at least $1-\alpha$. With such an oracle, the improvement in accuracy is as follows,
> > >
> > > $\Delta(M,m_x,\alpha) =\frac{1-\alpha}{m_x} - \frac{1}{M}=\frac{M(1-\alpha) -m_x}{m_xM} = \frac{M-m_x}{m_xM} -\frac{\alpha}{m_x} =\Delta(M,m_x) -\frac{\alpha}{m_x}$
> > >
> > > Note, here $m_x$ is a random variable, which depends on the effectiveness of the probabilistic oracle $\mathcal{P}$ and $\alpha$. Smaller set sizes $m_x$ and higher coverage probabilities $1-\alpha$ lead to larger gains in accuracy.
> > >
> > >
> > > Thus, even if we assume LLM as a random predictor, we should expect improvement in accuracy with CROQ, provided we can construct a probabilistic oracle $\mathcal{P}$, either using information from the same LLM or through external knowledge sources (such as other LLM, text embeddings, etc.). **Conformal prediction is a rigorous statistical framework using which one can construct such a probabilistic oracle.** Using CP in CROQ allows one to characterize the downstream accuracy improvements based on coverage level (1-$\alpha$) and set sizes of the probabilistic oracle.
> > >
> > > The alternative procedure based on directly tuning the "quality threshold" (say $\tau$) lacks such interpretation and analysis. Doing so brings it back to the conformal prediction framework, i.e., the quality threshold $\tau$ will correspond to a coverage of $1 -\alpha_\tau$ for some $\alpha_\tau \in [0,1]$.
> > >
> > >
> > > **3. Suggestion on spliting CP-OPT and CROQ into two papers**
> > >
> > > We have clarified the relationship between these methods, demonstrating that both are integral to our paper's goal of improving uncertainty quantification, accuracy, and deferral rates in MCQ tasks using LLMs.
> > >
> > > We hope our responses have addressed your concerns and would kindly ask you to reconsider your score. We are happy to answer any further questions you may have.

---

> ### Comment · Reviewer_pywK · 2024-12-02
>
> I thank the authors for the comprehensive answer. I would again like to make comments:
>
> **Regarding 1.**
>
> > In contrast, our work aims to improve post-hoc uncertainty quantification of LLMs for MCQ and tool-selection tasks.
>
> I am not convinced by this answer. From my understanding, for both MCQ and the tool selection task, one ends up with a setting that is almost identical to classification.
>
> **Regarding 2.**
>
> I thank the reviewers for this example. However, it is clear that this analysis only works for a random predictor, which is not interesting. In order for the argument to be convincing, the authors would have to demonstrate that such an analysis can be made for more relevant types of classifiers (ideally, any type of classifier). In general, I am afraid, it is straight-forward to see that no analysis or guarantees can be made. Hence, CROQ is an unnecessary detour in the general setting.
>
> All in all, I am still highly convinced that this paper must not be accepted. I will therefore keep my score with high confidence.

---

> ### Author Response · Authors · 2024-12-02
>
> **1. Regarding the MCQ vs. classification query**. These are fundamentally different problem settings. In classification, samples for each class share a common pattern, whereas, in MCQ tasks, there is no inherent pattern in the questions that determines which option is correct. For example, consider sentiment classification versus MMLU (MCQ). In sentiment classification, a sentence aligns with a specific label (e.g., "positive" or "negative"). However, in MCQ tasks, a legal question and a medical question might both have "A" as the correct answer, despite having entirely different features.
>
> We reiterate that, to the best of our knowledge, *CP-OPT is a novel contribution towards improving uncertainty quantification (UQ) and the accuracy of LLMs in finite response settings such as MCQ and tool-selection tasks.*
>
>
> **2. Analysis for better than random predictors.** In the previous example we chose to show the improvements for a random predictor as a worst-case scenario and to keep the example simple. However, it does not mean that the analysis or the guarantees do not extend to better than random predictors.
>
> **General Analysis**
>
> In general, consider a predictor (LLM) that has accuracy $a_k$ on questions with $k$ choices. It is fair to assume that as the number of choices $k$ decreases, the accuracy $a_k$ increases. This is also confirmed in our experiments (Figure 4). We refer to this as the *monotone accuracy property* of the predictor.
>
> Now, let the initial number of options in the questions be $M$ and after revising them with conformal prediction (CP) the questions have $m<M$ choices and it is guaranteed by CP that the true answer is still in the $m$ choices for $1-\alpha$ fraction of the questions. Then, the gain in accuracy after CROQ is as follows,
>
> $$\text{Gain} = \text{Accuracy After} - \text{Accuracy Before}$$
> The $\text{Accuracy After} = a_m$ times the fraction of questions for which true choice is in the revised question = $a_m (1-\alpha)$
>
>
> $$\Delta(M,m,\alpha) = a_m(1-\alpha) - a_M = (a_m - a_M) - \alpha a_m$$
> If $\alpha$ is fixed, then we should see improvements whenever $a_m > \frac{a_M}{1-\alpha}$. And if $\alpha$ is not fixed, then the gain $\Delta(M,m,\alpha) > 0$, **for any** $\alpha < \frac{a_m - a_M}{a_m}$. By the monotone accuracy property of the predictor $a_m - a_M >0$, that means any $\alpha \in (0,  \frac{a_m - a_M}{a_m})$ will yield a gain in accuracy.
>
> **Numerical Example**
>
> To instantiate it more clearly, Suppose LLM has 50% accuracy for questions with 10 options ($M=10$) and 60% accuracy for questions with 5 options ($m=5$) i.e. $a_{10} = 0.5$ and $a_5=0.6$. Suppose conformal prediction maps these 10 option questions to 5 option questions while ensuring that for 95% (i.e. $\alpha = 0.05$) of the questions, the true answer is still in the reduced set of 5 choices. With this, the new accuracy (after CROQ) will be,
>
> $$\text{New Accuracy} = a_5*(1-\alpha) = 0.6×0.95=0.57$$
> This means the new accuracy is 57%, which is an absolute improvement of 7% over the previous accuracy (i.e. before CROQ).
>
> The above analysis and example clearly demonstrate that the accuracy improvements with CROQ extend to a wide range of predictors, not just random ones. This highlights how the *use of conformal prediction (CP) in CROQ enables a systematic characterization of accuracy gains.*
>
> We sincerely hope the reviewer will consider our comprehensive responses and revisit their evaluation.

---

> > ### Comment · Reviewer_pywK · 2024-12-03
> >
> > I thank the authors for the response. I will only comment on the second point, since I have made clear my opinion on the first one.
> >
> > It is not clear at all to me why $\alpha_m - \alpha_M > 0$, in general. Furthermore, the authors are talking about accuracy improvement for *some* value of $\alpha$, for which I do not doubt that the method works. My question is, and this should be the headline of the manuscript, why I should perform this rather complicated algorithm to end up with something that could be done in a straightforward way. I believe that the authors must show how *guarantees* can be made for the downstream predictor, for *any* value (and depending on) $\alpha$. Otherwise, I simply see no point in this method.
> >
> > This is my final assessment and I recommend a clear rejection for the current state of the manuscript. I thank the authors for all their efforts.

---

> > > ### Author Response · Authors · 2024-12-03
> > > **CROQ vs. optimizing a quality threshold**
> > >
> > > We thank the reviewer for extensively engaging with us. We would like to offer one final point just for the record.
> > >
> > > We don't believe that CROQ is complicated or that it is fundamentally distinct from the reviewer's proposal to optimize a quality threshold. In both cases, the procedure looks like the following:
> > >
> > > 1. Fix a score function, which we could also call a quality function. This function evaluates how plausible a response option is with respect to a question. (In the present case, our function relies on the LLM itself, but as we emphasize in the paper, we can use any arbitrary function.)
> > > 2. Choose a grid of score function thresholds to evaluate.
> > > 3. Evaluate the chosen thresholds on a validation set: for each threshold and each question in the validation set, construct a set by thresholding the scores of the response options and query an LLM with the question and the chosen set of response options.
> > > 4. Choose the threshold that yields the highest accuracy.
> > >
> > > The conformal prediction perspective provides a **principled way to choose the candidate thresholds in step 2**. Under this perspective, we choose quantiles of the distribution of score function values on correct answers, which yield coverage guarantees for the resulting sets. Since the coverage values by definition lie in $[0, 1]$, we know we can span the whole space of meaningful values of the threshold. As the coverage approaches 0, the average set size will approach 0, and as the coverage approaches 1, the average set size will approach the number of response options.
> > >
> > > By contrast, suppose we heuristically chose a set of thresholds to evaluate in step 2. We might inadvertently choose thresholds which induce coverage values in a small range, say, 0.5 to 0.6. Then we'd be upper bounding downstream accuracy at 0.6 without even realizing it.
> > >
> > > Since evaluating empirical quantiles is computationally trivial, choosing the thresholds based on their coverage values in step 2 rather than by some other means does not add any meaningful overhead to the procedure.
> > >
> > > We note also that the conformal sets can be used in other downstream workflows in which knowing the coverage might be important. For example, suppose that the reduced answer sets were going to be passed to a human as opposed to an LLM. It would be useful for human decision makers to know how often they should expect to encounter sets that don't contain the correct answer option.
> > >
> > > In short, we think $\alpha$-tuning with CROQ is a simple and principled way to reduce the space of answer options when querying an LLM with a MCQ. The hallmark of this procedure is that rather than heuristically choosing candidate thresholds, it chooses the empirical quantiles of a score function, which are trivial to compute and which yield coverage guarantees for the resulting sets. When combined with CP-OPT, this procedure yields a favorable tradeoff between set size and coverage, which can improve final accuracy.

---

### Official Review · Reviewer_mTL6 · 2024-10-26

**Soundness:** 2
**Presentation:** 3
**Contribution:** 2
**Rating:** 5
**Confidence:** 4

**Summary:**

The paper proposes two techniques related to conformal prediction. For the first technique, the paper proposes a method for minimizing the size of the set of outputs in order to meet the coverage objective of the conformal predictor. As the objectives are non-differentiable, the paper proposes a differentiable approximation based on using the sigmoid to approximate the indicator function. For the second technique, the paper proposes a two stage method for improving the preformance of MCQ prediction. The method first use the conformal predictor to reduce the number of options to consider in the MCQ question, then predicts again with the remaining option.

**Strengths:**

The proposed method for reducing the size of the set of outputs is natural and relatively simple, which is good. The experimental results for the proposed two stage method for improving performance in MCQ questions are good.

**Weaknesses:**

The experimental results for reducing the size of the set of outputs did not show much improvement for the proposed method. In most of the cases considered (except 3 cases), the size is smaller but the coverage is also correspondingly smaller, so the improvement is not convincing. The reason for improvement in the two stage MCQ method is not clear and not much insights is provided in the paper on it. The analogy to Monty Hall is not convincing and is actually misleading as the mechanism for the improved performance in the two stage prediction method in Monty Hall is not present here. Note that no additional information is provided unlike in the Monty Hall case and if the correct posterior probability for each option is provided by the predictor, the optimal action is to predict the option with the highest probability, a one stage method.

**Questions:**

If possible, some insights on why improvement is small for the first technique would be useful. Similarly, insights of why the two stage method is helpful would be useful, if possible. Given that two stages is helpful, would even more stages be even more helpful?

---

> ### Author Response · Authors · 2024-11-23
> **Response to Reviewer mTL6**
>
> We thank the reviewer for their thoughtful feedback and constructive comments. We are pleased to hear that you found our methods natural and simple, and appreciated the experimental results demonstrating the effectiveness of our two-stage CROQ procedure. Below, we address your concerns in detail.
>
> ### **1. On set size reduction with CP-OPT**
>
> We acknowledge the reviewer's observation regarding the varying effectiveness of CP-OPT in reducing set sizes across different settings. We provide additional results (Table 10 and Figures 6-13) and insights to aid in understanding the effectiveness of CP-OPT.
>
>
>  **a. Average set size reduction.** In Tables 1 and 10 we see total 4 settings where CP-OPT reduces average set size significantly without losing on coverage and in rest of the settings (with a few exceptions) we see reduction in average set size with CP-OPT but at a slightly lower coverage than logits. In some of these settings we see logits were overcovering, thus bringing it down closer to 95\% is desirable. Moreover, minor variations in the coverage are anticipated due to calibration on finite samples.
>
>
>  **b. Distribution of set sizes.** To better understand the set sizes produced by CP-OPT and logit scores we visualize the histograms of their set sizes in Figures 5-11 for the settings in Table 1. In these figures we can see CP-OPT consistently reduces the proportion of large set sizes and increases the proportion of smaller set sizes. This redistribution explains the settings where CP-OPT leads to improvements in the CROQ (Tables 11, 12, 13, 18, and 19) and deferral procedures (Figure 3).
>
>
>  **c. Empirical factors.** CP-OPT is a principled method designed to learn the optimal scores for conformal prediction (CP). By design, these scores aim to produce smaller prediction sets compared to LLM logits, while maintaining the same coverage level. However, empirical performance depends on several factors, such as the quality of features used for score learning, the number of samples available for calibration, and the specifics of the training procedure.
>
>
>
> ### **2. Insights on improvements with CROQ**
>
> We agree with the reviewer that if the predictor initially provides the correct posterior probability for each answer option, then the optimal action is to select the option with the highest probability. In this case, since the answer to the MCQ is precisely defined in our approach as the option to which the LLM assigns the highest probability, the LLM would get the question right on the first round, so there would be nothing to gain from querying the LLM again.
>
> What we observe, however, is that reducing the set of available answers changes the probabilities that the LLM assigns to each answer option. Table 3 in the appendix illustrates that the highest probability answer changes approximately 3-15\% of the time after CROQ (summing the cases when it changes from correct to incorrect and vice versa). These changes are what produce the overall increase in accuracy that we generally observe with CROQ.
>
> ### **3. Regarding the analogy to Monty Hall**
>
> We appreciate the reviewer's perspective, and we here attempt to explain our intended meaning more clearly. The conformal set which is produced in the first stage of CROQ contains the correct answer with some user-specified probability, say 95%. In that sense, we imagine that an (imperfect) oracle is opening some number of doors (answer options) that with high probability reveal only goats (incorrect answers). Those answer options are eliminated, and then the LLM is queried again with the remaining set of answers. As described above, we see that in some proportion of cases, the LLM "decides to switch" its answer as a result of this procedure.
>
> The oracle's "knowledge" comes from the distribution of the scores of correct answers, so it is not contained within the LLM's representation of any given query or the probabilities it assigns to the answer options for that query. In that sense, it is extra information that gets added to each query. Additionally, the score function and the resulting distribution of scores can come from any source. Although in our experiments we used the same LLM both to generate conformal scores and to answer multi-choice questions, the conformal scores could come from another LLM, from embeddings that measure semantic similarity between questions and answers, etc. The extra information that the conformal procedure represents can therefore come entirely from an external source. (In future work we plan to run experiments in which we generate conformal scores externally.)
>
> We note that another reviewer expressed similar concerns and recognize that we did not explain our analogy convincingly. We hope that the above explanation clarifies our perspective. We have added similar explanatory language along these lines to the paper.

---

> > ### Comment · Reviewer_mTL6 · 2024-11-24
> >
> > Thank you for the response. I am still doubtful that provision of addition information like in the case of Monty Hall is the mechanism for improvement in your two stage process. There are other possible explanations, e.g. the LLM has been trained extensively to answer questions with a small number of choices but not with large number of choices, hence there is a distribution shift when it is tested with questions with large number of choices. If removing choices is easy to learn using a small dataset as done in conformal prediction, the two stage process that transforms problems with large number of choices into problems with small number of choices could be more effective. But this is speculation as well. Having clear understanding and evidence on the mechanism responsible for the improvement would strengthen the paper.

---

> ### Author Response · Authors · 2024-12-01
> **Response to Reviewer mTL6 (Part 1)**
>
> Thank you for the comment. We provide mathematical insights into why we expect CROQ to improve performance and empirical evidence **(Figure 4 and Tables 4-21)** showing that CROQ's improvements are consistent across a range of number of response options, suggesting the gains are due to the reduction in uncertainty.
>
>
> **1. Mathematical reasoning for CROQ is based on a simple fact that reduction in uncertainty can help even a random predictor.** This principle is also at play in Monty Hall (and thus the connection). We explain how this helps with a deterministic and a probabilisitc oracles. Consider a random black-box predictor, when given $M$ choices, it selects one option randomly and outputs it as the answer, i.e., its probability of correctness is $1/M$.
>
> a. If there is a **deterministic oracle** that can reduce the choices to $m<M$, while ensuring that the true answer is among the $m$ choices, then the probability of correctness of the same random predictor would be $1/m$. Implying accuracy improvement $\Delta(M,m) = \frac{1}{m} - \frac{1}{M} = \frac{M-m}{mM} > 0$. The smaller the $m$, the larger the improvement.
>
> b. In practice, we do not have such a deterministic oracle that can reduce the choices to $m$ while retaining the true answer. Suppose, instead we have a  **"probababilistic oracle"** $\mathcal{P}$ that reduces the initial set of $M$ choices (for a randomly drawn question $x$) to $m_x$ while ensuring that the true answer is in the selected $m_x$ choices with probability at least $1-\alpha$. With such an oracle, the improvement in accuracy is as follows,
>
> $\Delta(M,m_x,\alpha) =\frac{1-\alpha}{m_x} - \frac{1}{M}=\frac{M(1-\alpha) -m_x}{m_xM} = \frac{M-m_x}{m_xM} -\frac{\alpha}{m_x} =\Delta(M,m_x) -\frac{\alpha}{m_x}$
>
> Note, here $m_x$ is a random variable, which depends on the effectiveness of the probabilistic oracle $\mathcal{P}$ and $\alpha$. The gain approaches $\Delta(M,m)$ as $m_x \to m$ and $\alpha \to 0$. In other words, if $\mathcal{P}$ outputs a small subset (i.e., small $m_x$) with high coverage probability $1-\alpha$ (i.e., small $\alpha$), then we will have a higher gain in accuracy.
>
>
> The above arguments show, in principle, that reducing uncertainty can help improve the accuracy of even a random predictor. Thus, even if we assume LLM as a random predictor, we should expect improvement in accuracy with CROQ, provided we can construct a probabilistic oracle $\mathcal{P}$, either using information from the same LLM or through external knowledge sources (such as other LLM, text embeddings, etc.) In our experiments, we see that LLM predictions are better than random, and we can construct $\mathcal{P}$ using the conformal prediction and information from the same LLM, resulting in improvements in accuracy.
>
>
> **2. Empirical results showing the accuracy improvement is due to a reduction in the uncertainty.**
>
> We have run CROQ with a groundtruth oracle to demonstrate that it is the reduction in uncertainty that helps LLM answer the revised question with higher accuracy. **Please see Figure 4 in the Appendix (page 14)**. In this experiment, we use the Truthful QA dataset with 15 response options. We first construct conformal prediction sets using logit scores. With these prediction sets, we then leverage groundtruth knowledge to reduce the prediction set size by 0 to 10 options while ensuring that coverage remains constant. We can clearly see in Figure 4, the accuracy of the LLM after requerying increases as more choices are eliminated (smaller prediction set). These results are consistent with the above mathematical arguments on accuracy improvement with reduction in uncertainty and also motivate the use of a score function optimization (CP-OPT) to reduce uncertainty (minimize set sizes) while controlling coverage.
>
> Further, we have extensive experiments on settings with 10 and 15 response options. Please see Tables 4 to 21 in the Appendix; we have also included Table 9 in the comment for your reference. We see CROQ improves accuracy in a vast majority of the settings. The tables provide improvements conditioned on set sizes, i.e., the number of response options in the revised question. As we can see, the improvements are not restricted to only small set sizes, suggesting the improvements are likely due to a reduction in the uncertainty.
>
> We reiterate our focus is on validating the hypotheses (H1, H2, H3). We have provided mathematical insights into why we expect these hypotheses to be true and provided extensive empirical evidence to validate them. Our work shows LLMs can self-correct with CROQ, and it would be exciting future work to develop a precise mechanistic understanding of it.
>
>
> We hope the above mathematical reasoning and empirical results in Figure 4, and the attached table address your concerns. We are happy to answer any further queries you may have.

---

> ### Author Response · Authors · 2024-12-01
> **Response to Reviewer mTL6 (Part 2)**
>
> **Part of Table 9 with Llama-3 model and Logit scores**
>
>
>
> | **Set Size**    | **1** | **2**   | **3**     | **4**     | **5**     | **6**     | **7**     | **8**     | **9**     | **10** | **Overall**
> |------------|------------|----------------|----------------|----------------|----------------|----------------|----------------|----------------|----------------|-------------|------------------|
> | **Coverage**    | 94.73      | 91.44          | 91.47          | 94.96          | 95.29          | 96.44          | 96.88          | 97.18          | 98.01          | 100.00      | 95.57            |
> | **Fraction**    | 16.67      | 11.51          | 9.04           | 8.24           | 7.81           | 8.01           | 8.75           | 9.67           | 8.96           | 11.33       | 100.00           |
> | **Acc. Before** | 94.73      | **78.14** | 62.99          | 52.88          | 50.00          | 40.74          | 39.76          | 34.85          | 33.91          | 30.47       | 55.35            |
> | **Acc. After**  | 94.73      | 77.22          | **65.22** | **57.35** | **51.37** | **45.04** | **40.71** | **37.55** | **34.70** | 30.47       | **56.68***  |
>
>
>
> **Part of Table 9 with Llama-3 model and Ours (CP-OPT) scores**
>
>
> | **Set Size**    | **1** | **2**   | **3**     | **4**     | **5**     | **6**     | **7**     | **8**     | **9**     | **10** | **Overall**
> |------------|------------|----------------|----------------|----------------|----------------|----------------|----------------|----------------|----------------|-------------|------------------|
> | **Coverage**    | 94.61      | 92.23          | 90.39          | 92.82          | 95.85          | 96.66          | 96.88          | 97.46          | 99.60          | 100.00      | 95.02            |
> | **Fraction**    | 14.76      | 12.38          | 11.49          | 10.57          | 11.43          | 11.00          | 9.52           | 7.95           | 5.95           | 4.95        | 100.00           |
> | **Acc. Before** | 94.61      | 80.54          | 63.22          | 50.06          | 47.04          | 39.48          | 37.53          | 31.19          | **29.14** | 27.34       | 55.35            |
> | **Acc. After**  | 94.61      | **81.02** | **63.95** | **54.32** | **52.54** | **42.72** | **40.15** | **32.99** | 28.14          | 27.34       | **57.26***  |
>
>
> **Part of Table 9 with Phi-3 model and Logit scores**
>
> | **Set Size**    | **1** | **2**   | **3**     | **4**     | **5**     | **6**     | **7**     | **8**     | **9**     | **10** | **Overall**
> |------------|------------|----------------|----------------|----------------|----------------|----------------|----------------|----------------|----------------|-------------|------------------|
> |**Phi-3**     | 95.25      | 92.20          | 91.24          | 92.83          | 95.32          | 96.40          | 96.21          | 94.89          | 97.74          | 100.00      | 94.74            |
> | **Fraction**    | 17.23      | 13.24          | 10.56          | 10.92          | 10.40          | 9.57           | 9.09           | 7.67           | 5.77           | 5.55        | 100.00           |
> | **Acc. Before** | 95.25      | 79.48          | 62.36          | 55.43          | 46.92          | 45.78          | 41.64          | 33.75          | 31.89          | 27.78       | 58.59            |
> | **Acc. After**  | 95.25      | **81.81** | **67.42** | **61.30** | **53.42** | **48.39** | **42.95** | **34.83** | **32.7** | 27.78       | **61.25***  |
>
> **Part of Table 9 with Phi-3 model and Ours (CP-OPT) scores**
> | **Set Size**    | **1** | **2**   | **3**     | **4**     | **5**     | **6**     | **7**     | **8**     | **9**     | **10** | **Overall**
> |------------|------------|----------------|----------------|----------------|----------------|----------------|----------------|----------------|----------------|-------------|------------------|
> | **Coverage**    | 94.81      | 90.79          | 91.27          | 92.95          | 94.73          | 95.58          | 95.77          | 97.28          | 98.52          | 100.00      | 94.53            |
> | **Fraction**    | 19.19      | 13.02          | 10.47          | 10.43          | 10.59          | 9.39           | 8.41           | 7.43           | 6.40           | 4.68        | 100.00           |
> | **Acc. Before** | 94.81      | 77.12          | 64.17          | 54.38          | 47.31          | 44.50          | 39.21          | 32.11          | 29.87          | 25.38       | 58.59            |
> | **Acc. After**  | 94.81      | **79.40** | **68.14** | **60.41** | **54.71** | **48.93** | **39.77** | **32.43** | **31.35** | 25.38       | **61.30***  |

---

> > ### Author Response · Authors · 2024-12-02
> >
> > Dear Reviewer,
> >
> > As we near the end of the discussion period, we would appreciate it if you could review our response and let us know if there are any remaining questions we can answer.
> >
> > Thank you!

---

> > > ### Comment · Reviewer_mTL6 · 2024-12-03
> > >
> > > Thanks for the additional clarification. I accept that reducing the number of options by using an appropriate threshold to remove lower ranked ones improve performance experimentally. However, the mechanism for why the LLM should sometimes change it's answer in the second round remains unclear. The essence of the Monte Hall problem is the analysis of the mechanism of why it is better to change the answer in the second round, and the mechanism does not seem to apply here?

---

> > > > ### Author Response · Authors · 2024-12-03
> > > >
> > > > We are glad that our previous response helped in clarifying some of the queries. Our response to the remaining questions is as follows,
> > > >
> > > > First, we would like to clarify that *our reference to the Monty Hall problem is intended only as a conceptual analogy to motivate CROQ*, rather than suggesting that LLMs emulate the switching strategy (as in Monty Hall) to improve accuracy. Just as reducing the number of choices in Monty Hall improves the player’s chances of winning, we expect reducing the number of options in CROQ would improve the LLM’s likelihood of answering correctly.
> > > >
> > > > However, there are important distinctions. Unlike in the Monty Hall problem, the LLM is not a random predictor and it has some baseline accuracy, so *for LLM  always switching from the initial choice may not be the best strategy in this case*. However, LLMs do switch their answer for *some questions* (not for all) when the number of options are reduced. Intuitively, a question with a large number of options has more noise (distractor options), and eliminating some of the noisy ones could help LLM identify the correct option (thus switching).
> > > >
> > > > We can characterize this using the *monotone accuracy property* – we say a predictor has this property when its accuracy increases monotonically as the number of choices decreases. Our empirical results in Figure 4 suggest that LLMs are likely to have this property. Understanding why LLMs exhibit this property would be interesting future work, and here we show that whenever a predictor has this property, we can expect to see accuracy improvements with CROQ.
> > > >
> > > > Consider a predictor (LLM) that has accuracy $a_k$ on questions with $k$ choices.  Now, let the initial number of options in the questions be $M$ and after revising them with conformal prediction (CP) the questions have $m<M$ choices and it is guaranteed by CP that the true answer is still in the $m$ choices for $1-\alpha$ fraction of the questions. Then, the gain in accuracy is as follows,
> > > >
> > > > $$\text{Gain} = \text{Accuracy After} - \text{Accuracy Before}$$
> > > >
> > > > The accuracy after = $a_m$ times the fraction of questions for which true choice is in the revised question = $a_m (1-\alpha)$
> > > >
> > > > $$\Delta(M,m,\alpha) = a_m(1-\alpha) - a_M = (a_m - a_M) - \alpha a_m$$
> > > >
> > > > If $\alpha$ is fixed, then we should see improvements whenever $a_m > \frac{a_M}{1-\alpha}$. And if $\alpha$ is not fixed, then the gain $\Delta(M,m,\alpha) > 0$, for any $\alpha < \frac{a_m - a_M}{a_m}$. By the monotone accuracy property of the predictor $a_m - a_M >0$, that means any  $\alpha \in (0, \frac{a_m - a_M}{a_m} )$ will yield a gain in accuracy.
> > > >
> > > > For example, say LLM has 35% accuracy on questions with 20 options and 50% accuracy on questions with 10 options. Now, say we have 1000 questions with 20 options. With a single round procedure, we will get 35% accuracy on these questions.
> > > > Now, suppose conformal prediction reduces the choices from 20 to 10 for each question and ensures  90% coverage i.e., for 900 of the questions, the true answer choice is still in the remaining 10 choices. As LLM has 50% accuracy on the 10 choices questions, the new accuracy of the original questions will be $0.90*0.5$ = 45%, i.e., a 10% improvement.
> > > >
> > > > We hope this clarifies your question and addresses any remaining concerns. We would sincerely appreciate it if you could review our overall responses and consider updating your scores

---

### Official Review · Reviewer_yss3 · 2024-11-03

**Soundness:** 3
**Presentation:** 3
**Contribution:** 2
**Rating:** 6
**Confidence:** 3

**Summary:**

The paper considers the setting where LLMs are used to answer MCQ-style problems. The authors consider the use of conformal prediction in this setting and make two methodological contributions:
1. Firstly, the authors propose a method of optimising score function which leads to smaller confromal sets on average.
2. Secondly, the authors investigate what happens if the LLMs are provided revised questions where the options are restricted to the conformal sets.

**Strengths:**

1. The paper is well-written
2. The methodology is clearly explained
3. The paper includes extensive experiments to empirically investigate the methodology proposed.

**Weaknesses:**

1. The paper considers the MCQ-setting, which could be somewhat restrictive. Is it possible to extend these ideas to settings where the model is not provided with options?
2. My main concerns regarding this paper are two fold:

i. Firstly, the methodologies proposed are somewhat orthogonal. The CP-OPT is a general methodology for optimising the score function which could be applicable to any CP problem (and is not specific to LLMs). In comparison, the CROQ methodology simply re-queries the model using the conformal sets. These methodologies are completely independent of each other and therefore I think the main contribution of the paper is not very coherent.

ii. Secondly, the methodologies proposed themselves are not very novel. For example, Cherian et al., 2024 (which the authors cite) seem to propose a very similar methodology of optimising the score functions as that proposed in this paper. Can the authors please elaborate on how their methodology is different? Similarly, the idea of re-querying the model does not seem very novel either.

3. In Figure 3, please also add the uncertainty in the accuracies for all methods. In most cases, its unclear whether CP-OPT produces a better accuracy than logits.

4. While it can be seen that CP-OPT leads to smaller sets on average, it is not convincing from the empirical results that the accuracy of revised questions is strictly better when using CP-OPT as opposed to just logits. Can the authors explain why CP-OPT does not seem to do better than logits at CROQ?

5. Can the authors explain why for MMLU there is no difference in Figure 4 between CP-OPT and logit methods whereas for Truthful QA, CPT-OPT leads to fewer deferrals and higher accuracy?

**Questions:**

See weaknesses section above.

---

> ### Author Response · Authors · 2024-11-23
> **Response to Reviewer yss3 (Part 1)**
>
> Thank you for the detailed and constructive feedback. We appreciate your positive assessment of the paper's presentation, experiments, and contributions. Our response to the concerns is as follows:
>
> ### **1. On MCQ-setting**
> First, we emphasize that the MCQ (or finite response options) setting is a fairly broad setting that covers several use cases of interest, e.g., tool selection (Tang et al., 2023; Qu et al., 2024) and question answering (Kumar et al., 2023; Su et al., 2024). Moreover, popular benchmarks, e.g., MMLU (Hendrycks et al., 2021), to evaluate language understanding of LLMs, are based on MCQ. In general, the scenarios where LLMs have to select from a finite number of responses can be expressed as MCQs, and our framework will be directly applicable to those settings. Second, there are a few works on using conformal prediction for LLMs in open-ended QA (e.g., Mohri et al., 2024), so the ideas presented in our paper can potentially be extended to open-ended response settings. Exploring this would be an interesting direction for future work.
>
> ### **2. Perceived orthogonality of our methods**
> We agree that CP-OPT and CROQ can function independently to reduce uncertainty and improve accuracy in LLMs. However, when used together, they complement each other effectively, serving the common goal of reducing uncertainty in LLM outputs. CP-OPT refines prediction sets by minimizing their sizes while maintaining coverage guarantees, and CROQ utilizes these refined sets to reduce the number of answer options presented to the LLM, improving its performance. The motivation for integrating CP-OPT with CROQ is supported by evidence in Figure 4, which demonstrates that as prediction set size decreases (simulated using ground truth), CROQ's accuracy consistently improves.
>
> To make this connection explicit, we have added Hypothesis 3 (H3) to the paper, which evaluates the combined effectiveness of CP-OPT and CROQ. Specifically, H3 examines whether CROQ with CP-OPT scores outperforms CROQ with logits. The hypothesis is supported by empirical results showing that CP-OPT generally leads to better accuracy in CROQ, reinforcing the complementary nature of these methodologies. While there are cases where the improvements are marginal, the overall findings underscore the advantage of using CP-OPT scores in CROQ, demonstrating their synergy in improving decision-making with LLMs.
>
> The loose coupling between CP-OPT and CROQ is by design. CP-OPT can be used generally to reduce set sizes in conformal prediction, and CROQ can work with any score function, including CP-OPT and LLM logits.
>
> ### **3. Novelty**
>
> **a. CP-OPT.** It is designed to address the need for principled score functions in decision-making settings such as MCQs, where prior works have relied on either model logits or heuristic scores (Kumar et al., 2023; Su et al., 2024). Cherian et al., 2024, focus on factuality guarantees in open-ended generation tasks. In their setting, there is not necessarily a single correct response, so the notion of coverage is redefined around acceptability or factuality rather than correctness. In contrast, CP-OPT aims to generate the smallest possible set of response options while ensuring that the correct option is included with high probability, adhering to the coverage guarantee based on correctness.
>
> **b. CROQ.** While re-querying a model might seem conceptually straightforward and there could be several heuristic strategies to prompting and re-querying, CROQ is a *unique and principled framework* leveraging conformal prediction (CP) to create a refined question. Since CROQ revises the question with the options in the prediction set obtained from CP, it ensures that the correct answer remains in the revised question with high probability, due to the coverage guarantee of CP, which is distribution- and model-agnostic. Our experiments show that re-querying with CROQ consistently improves accuracy.
>
> **c. Combining CP-OPT and CROQ.**  Together, CP-OPT and CROQ form a coherent pipeline where refined prediction sets from CP-OPT are used in CROQ to obtain a refined question. Our experiments show that running CROQ with CP-OPT is a better choice than running it with logits.
>
> ### **4. Uncertainties and significance of accuracies in Figure 3 (now Figure 2)**
>
> The goal of this figure is to show how CROQ reacts to variations in the coverage parameter $\alpha$. We elaborate on the differences between CP-OPT and logits in CROQ in point 5 below and refer to the added discussion on Hypothesis 3 (H3) in the updated paper. Running the CROQ procedure for all $\alpha$ is computationally demanding, so unfortunately, we cannot provide them in the rebuttal period; however, we will include them in the camera-ready version.
>
> [Response continues in the next comment]

---

> ### Author Response · Authors · 2024-11-23
> **Response to Reviewer yss3 (Part 2)**
>
> ### **5. CP-OPT vs logits on CROQ**
>
> The impact of CP-OPT on CROQ accuracy depends on the extent of set size reduction. To further investigate this, we conducted additional experiments specifically addressing Hypothesis 3 (H3) in the paper, which evaluates whether CROQ with CP-OPT scores outperforms CROQ with logits.
>
> The results, as presented in Tables 11, 12, 13, 14, 17, 18, and 19, align with our expectation that CP-OPT improves accuracy when the set size reduction is substantial. For example, in TruthfulQA with 10 options, CP-OPT leads to significant gains by effectively refining uncertainty. However, in cases where the reduction in set size is minimal (e.g., Tables 4, 5, 8, 9, 15, and 16), the accuracy improvement is less pronounced. This highlights that the benefit of CP-OPT is most evident in scenarios with larger or more uncertain initial prediction sets.
>
> Overall, we see that CP-OPT generally enhances CROQ performance, and the magnitude of improvement varies depending on dataset and task characteristics.
>
> ### **6. Results on TruthfulQA and MMLU in Figure 4**
> In Figure 4, CP-OPT leads to fewer deferrals in the TruthfulQA setting compared to logits, but in the MMLU setting, we do not see such a difference. This is due to differences in the distributions of the sets produced by these methods in the above settings. We have included histograms (distributions) of the set sizes in Figures 8(b) and 11(b) for MMLU and Truthful QA settings respectively. For a method to lead to fewer deferrals, it should have lower mass on larger set sizes and consequently higher mass on smaller set sizes. We see clear evidence for this in the TruthfulQA setting, but in the MMLU setting, the reductions on large sets are small, leading to nearly similar performance with logits on the deferral task.
>
> ### **References**
>
> 1. Hendrycks et al., 2021, *Measuring massive multitask language understanding*.
>
> 2. Kumar et al., 2023, *Conformal prediction with large language models for multi-choice question answering*.
>
> 3. Su et al., 2024, *Conformal prediction for large language models without logit-access*.
>
> 4. Qu et al., 2024, *Tool learning with large language models: A survey*.
>
> 5. Tang et al., 2023, *Toolalpaca: Generalized tool learning for language models with 3000 simulated cases*.

---

> > ### Comment · Reviewer_yss3 · 2024-11-24
> >
> > Firstly, I want to thank the reviewers for their response and for the changes made to the paper.
> >
> > 1. Regarding the orthogonality of the two methodologies (and H3), I am still not convinced regarding the authors' claim that "when used together, they complement each other effectively". For example, in Table 2, the use of CP-OPT instead of logits seems to even worsen the accuracy of the models in some cases. As the authors pointed out, the combined effectiveness of these two methods seems highly data-dependent. While it might improve performance on some datasets, it could also worsen the performance on others.
> >
> > 2. I am also not convinced by the novelty of CR-OPT as compared to Stutz et al., 2022 or Cherian et al., 2024. While the application explored might be different in Cherian et al., 2024, the key aspects of optimising the score function still seem quite similar.

---

> ### Author Response · Authors · 2024-12-01
> **Response to Reviewer yss3 (Part 1)**
>
> Thank you for the comment. We provide clarifications on the two points that you mentioned,
>
> **1. Effectiveness of CP-OPT in CROQ**
>
> We summarize our results over all the 21 settings considered in the paper (see tables in the comment). We see **in 16 (out of 21) settings, using CP-OPT leads to higher accuracy than using logits in CROQ**. In 1 of the settings, we do not see improvements, and in 4, there is a slight drop in accuracy with CP-OPT. Moreover, most of the times, we see numerically larger improvements than the drop in the 4 cases. These results provide substantial evidence on the effectiveness of using CP-OPT in CROQ. The cases with a drop in accuracy are likely an artifact of the empirical procedure based on finite samples thus, if we consider the relative improvements in the range $(-1,1)$ as insignificant, we can conclude in 9/21 settings, CROQ with CP-OPT does significantly better and in other cases it performs similar to logits.
>
>
>
> In the tables below, $a_0$= accuracy after CROQ with logit scores and $a_1$ = accuracy after CROQ with CP-OPT scores.
>
> **Results on CROQ with CP-OPT and logits (part 1 of the table)**
>
>  |                                            |         |         |         |        |         |         |         |         |         |
> |:------------------------------------------:|:-------:|:-------:|:-------:|:------:|:-------:|:-------:|:-------:|:-------:|:-------:|
> | **Model**                                  | Llama-3 | Llama-3 | Llama-3 | Phi-3  | Phi-3   | Phi-3   | Gemma-2 | Gemma-2 | Gemma-2 |
> | **Dataset**                                | MMLU-4  | MMLU-10 | MMLU-15 | MMLU-4 | MMLU-10 | MMLU-15 | MMLU-4  | MMLU-10 | MMLU-15 |
> | **Improvement ($a_1 - a_0$)**                    | -0.17   | **0.58**    | **0.51**    | **0.33**   | **0.05**    | -0.17   | **1.86**    | **4.00**       | **0.73**    |
> | **Relative Improvement $(a_1 - a_0)/a_0$** | -0.27   | **1.02**    | **0.94**    | **0.48**   | **0.08**    | -0.29   | **2.75**    | **7.42**    | **1.44**    |
>
>
> **Results on CROQ with CP-OPT and logits (part 2 of the table)**
>
> |                                            |              |               |               |              |               |               |
> |:------------------------------------------:|:------------:|:-------------:|:-------------:|:------------:|:-------------:|:-------------:|
> | **Model**                                  | Llama-3      | Llama-3       | Llama-3       | Phi-3        | Phi-3         | Phi-3         |
> | **Dataset**                                | TruthfulQA-4 | TruthfulQA-10 | TruthfulQA-15 | TruthfulQA-4 | TruthfulQA-10 | TruthfulQA-15 |
> | **Improvement ($a_1 - a_0$)**                    | **2.02**         | **1.78**          | **1.01**          | **0.25**         | **1.02**          | **3.29**          |
> | **Relative Improvement $(a_1 - a_0)/a_0$** | **3.63**         | **4.42**          | **2.54**          | **0.36**         | **1.92**          | **6.56**          |
>
>
>  **Results on CROQ with CP-OPT and logits (part 3 of the table)**
>
> |                                            |              |               |               |              |               |               |
> |:------------------------------------------:|:------------:|:-------------:|:-------------:|:------------:|:-------------:|:-------------:|
> | **Model**                                  | Llama-3      | Llama-3       | Llama-3       | Phi-3        | Phi-3         | Phi-3         |
> | **Dataset**                                | ToolAlpaca-4 | ToolAlpaca-10 | ToolAlpaca-15 | ToolAlpaca-4 | ToolAlpaca-10 | ToolAlpaca-15 |
> | **Improvement ($a_1 - a_0$)**                    | 0            | **0.24**          | -0.7          | **0.46**         | -0.35         | **0.46**          |
> | **Relative Improvement $(a_1 - a_0)/a_0$** | 0            | **0.27**          | -0.78         | **0.49**         | -0.39         | **0.51**          |

---

> ### Author Response · Authors · 2024-12-01
> **Response to Reviewer yss3 (Part 2)**
>
> **2. Novelty of CP-OPT**
>
> We have clarified the novelty of our work with respect to Stutz et al. (2022) and Cherian et al. (2024), both in our paper and in the previous response. We hope to clarify it further here,
>
> The procedure in **Stutz et al. (2022)** is applied at training time, aiming to improve the classifier's training so that the softmax outputs are better tailored for conformal prediction. **Cherian et al. (2024)** (Section 3.3) extend the ideas from Stutz et al. for *post-hoc* learning of scores, focusing on a *conditional coverage guarantee defined on factuality*.
>
> In contrast, **CP-OPT** performs *post-hoc optimization* of **set sizes, subject to a marginal coverage guarantee defined on correctness**. This approach is **specifically designed to improve uncertainty quantification for LLMs in MCQ tasks, a gap not addressed in prior works in these settings**.
>
> Empirically, we evaluate CP-OPT across three LLMs and datasets on MCQ and tool selection tasks, with variations in the number of options. We demonstrate the efficacy of CP-OPT in **reducing set sizes**, **improving accuracy in CROQ**, and lowering the number of high-uncertainty points, thereby **reducing the number of deferrals**.
>
> To the best of our knowledge, these are novel contributions towards improving UQ and the accuracy of LLMs in finite response settings such as MCQ and tool-selection tasks.
>
>
> We hope our response addresses your concerns on the efficacy of CP-OPT in CROQ and the novelty of CP-OPT. We are happy to answer any further questions you may have.

---

> > ### Comment · Reviewer_yss3 · 2024-12-02
> >
> > Thank you for the additional clarifications.
> >
> > Given the authors' response summarising the empirical results and methodological novelty, I am happy to increase my score. I would suggest including these clarifications in the paper as well.

---

> > > ### Author Response · Authors · 2024-12-02
> > >
> > > Thank you for taking the time to review our responses. We are glad to have addressed your queries and appreciate you increasing the score. We will make sure to include the clarifications in the paper.

---

### Official Review · Reviewer_T1gy · 2024-11-04

**Soundness:** 3
**Presentation:** 3
**Contribution:** 3
**Rating:** 8
**Confidence:** 4

**Summary:**

The paper presents a method where conformal prediction can be used to quantify and reduce output uncertainty for decision-making problems using LLMs.
Authors show that using standard LLM softmax logits in the case of MCQ can be improved more using their proposed conformal prediction scoring under coverage constraint CP-OPT in cases when LLM logits are not as highly informative. Further they propose CROQ (which is reprompting the LM with the previous conformal set produced) which are shown to increase accuracy further.

Empirically the proposed method seems to be promising when LLM softmax logits lacks information and the utilisation of CROQ

**Strengths:**

The paper is well written the problem is well motivated and presented well.
Overall the intuition behind using conformal prediction in the context of MCQ answer using LLMs is well appreciated.

Extensive experiments performed across 3 different MCQ datasets for different configurations for testing the utility of proposed conformal prediction set scoring framework CP-OPT (e.g. Fig 4. Table 1 , Fig 3).

Explaination of Hypothesis w.r.t empirical evidence for 4.2 is well presented.

**Weaknesses:**

- Limited baseline model coverage. It would have been good to see a couple more open source models, to test the gaps across further models with respect to Logits procedure.
- The design principles for learning CP-OPT seems limited as to using a 3-layer neural network. Some more additional details will be useful.

**Questions:**

Section 3.1.2 - cardinality of $\mathcal{D}_{train}$ is $n_t$ but in equation it shows $n$.

Fig 3: MMLU-10 and MMLU-15 for LLama, is there any specific reason as to there is a continued diminishing effect to revision as coverage parameter is increased as compared to other cases where the spike is less spread.
Although this is not the focus of the paper, any discussion on how this can be in some sense extrapolated to open ended question generation settings as well.

---

> ### Author Response · Authors · 2024-11-14
> **Initial response for clarification**
>
> Dear reviewer,
>
> Thank you for your thoughtful feedback and constructive comments on our paper. We will provide a full response to all points shortly. We wanted to seek a quick clarification on the additional experiments due to time constraints.
>
> Regarding your suggestion to run experiments on more models, we would be grateful if you could specify any particular models you would like to see included and the specific insights you hope these models will reveal.
>
> Since experiments with large language models are both time-intensive and costly, early guidance on model selection would be greatly helpful as we assess and prioritize what we can provide within the rebuttal period, so that we can engage in as constructive a discussion as possible.
>
>
>
> Thanks,
>
> The authors.

---

> ### Author Response · Authors · 2024-11-23
> **Response to Reviewer T1gy**
>
> Thank you for the thoughtful feedback and constructive comments on our paper. We are pleased to hear that you found our work well-motivated, well-written, and empirically promising for improving uncertainty quantification in LLMs in MCQ settings. Below, we address your questions and concerns in detail:
>
>
>
> ### **1. Evaluation on more language models**
>
> We evaluated our methods on `gemma-2-9b-it-SimPO` model (Meng et al., 2024) and the MMLU dataset with 4, 10, and 15 response options. Please see Tables 10, 11, 12, and 13 in the Appendix for the results. To aid in understanding and how they are used together, we have explicitly included Hypothesis 3 (H3). The results in this setting are consistent with our expectations and even more substantial than other settings for hypotheses H1 and H3. We discuss them below:
>
>  **H1. Set size reduction with CP-OPT.**
> In Table 10, we clearly see that CP-OPT reduces the average set size significantly while maintaining a similar coverage level. Moreover, in Tables 11, 12, and 13, CP-OPT reduces the fraction of points with larger set sizes and increases the fraction of points with smaller set sizes. For example, in Table 11, logit scores yield set size 15 for 41.7% of points, but CP-OPT reduces this to 25.96%.
>
>  **H2. CROQ with logit scores improves accuracy relative to baseline.**
> We observe small improvements in overall accuracy and significant improvements on points where logits produced smaller sets. While we expect CROQ to improve overall accuracy significantly, we find that logits have a higher proportion of points with large set sizes, meaning there is no substantial reduction in the uncertainty in a large portion of the revised questions. This explains the results on the overall accuracy. These results also highlight the unreliability of logits and how they can be a bottleneck in CROQ.
>
> **H3. CROQ with CP-OPT scores performs better than CROQ with logit scores.**
> By design, CP-OPT scores minimize set sizes while preserving the coverage guarantee. Thus, using these scores in the CROQ procedure should lead to a large portion of questions having lower uncertainty (fewer response options) after revision, and, conditional on the correct answer appearing in the revised question, we expect LLMs to be more likely to answer correctly if there are fewer response options in the revised question. The results in Tables 11, 12, and 13 align with this expectation. We see that running CROQ with CP-OPT scores results in higher accuracy than running it with logits.
>
>
> ### **2. Choice of the score function $\mathcal{G}$ in CP-OPT**
>
> We emphasize that our CP-OPT framework for learning the score function is general and can work with any reasonable function class $\mathcal{G}$. Since CP-OPT is a post-hoc procedure, meaning it operates on an already-trained LLM, we aim to use a sufficiently flexible class $\mathcal{G}$ that is not computationally intensive to train. For our experiments, we chose 3-layer neural networks for score learning and used them consistently. We have added more details about this in Appendix B.1.
>
>
>
> ### **3. Notation**
> Thank you for noting the inconsistency in $n_t$ and $n$. The $n$ in these equations should be $n_t$. We have updated this in the paper.
>
>
>
> ### **4. Diminishing effect in Figure 3 (now Figure 2)**
> In this figure, we show results with varying coverage parameter $\alpha$. Recall that a coverage of $1-\alpha$ means that $1-\alpha$ fraction of the prediction sets from CP contains the true answer choice. As $\alpha$ increases, the coverage ensured by the CP procedure decreases, meaning that with larger $\alpha$, a larger portion of revised questions will not contain the true answer, making it less likely for the LLM to provide the correct answer.
>
> Conversely, keeping $\alpha$ too small does not give a meaningful reduction in the noisy choices, so we do not see much improvement with smaller $\alpha$ either. In practice, this parameter can be tuned for a desired level of coverage and accuracy.
>
>
> ### **5. Extension to open-ended QA**
> There are a few works on using conformal prediction for LLMs in open-ended QA (e.g., Mohri et al., 2024). Extending our ideas to open-ended question answering (QA) would be an interesting direction for future work.
>
>
>
> ### **References**
>
> 1. Meng et al., 2024, *SimPO: Simple Preference Optimization with a Reference-Free Reward*
>    [https://arxiv.org/pdf/2405.14734](https://arxiv.org/pdf/2405.14734)
>
> 2. Mohri et al., 2024, *Language Models with Conformal Factuality Guarantees*
>    [https://arxiv.org/pdf/2402.10978](https://arxiv.org/pdf/2402.10978)

---

> ### Comment · Reviewer_T1gy · 2024-11-25
> **Response to authors**
>
> I appreciate the authors response to this rebuttal discussion and providing clarifications.
>
> The new experiments done on gemma-2-9b-it-SimPO seem promising from the results presented in Appendix Table 10-12 and seems to be in alignment with the previous hypothesis.
>
> I would like to keep my score unchanged but am positively inclined towards acceptance.

---

> > ### Author Response · Authors · 2024-12-01
> > **Response to Reviewer T1gy**
> >
> > Thank you for reviewing our clarifications and new experimental results. **The new results are consistent with our expectations and clearly demonstrate the efficacy of CP-OPT in set size reduction and accuracy improvement when CROQ is run with CP-OPT scores**.
> >
> > We’re glad to have addressed your concerns. If you have any further questions, we’d be happy to answer them and would appreciate it if you might reconsider updating your score.

---

> > > ### Comment · Reviewer_T1gy · 2024-12-02
> > > **Response to authors**
> > >
> > > After carefully reviewing through additional rebuttal comments on the new experiments and other reviewer's rebuttal responses justifying clarity in the draft and addressing it well, I am positively inclined towards accepting the paper and have increased my score.

---

> > > > ### Author Response · Authors · 2024-12-02
> > > >
> > > > We’re glad to have addressed your queries and appreciate you taking the time to review our responses to other reviewers as well and updating your score.

---

### Author Response · Authors · 2024-11-23
**Common Response (Part 1)**

# Common Response

We thank all of the reviewers for their insightful and positive feedback. We have used their suggestions to improve our draft, added experiments, and improved the clarity of our work. Before providing individual responses, we (1) summarize the strengths highlighted by reviewers, (2) address major common concerns, and (3) describe new experiments that strengthen our work.




### **Strengths**


**1. Novel, interesting, and simple methods (T1gy, mTL6, pywK)**

Reviewers liked the simplicity and novelty of our methods CP-OPT (for reducing prediction set sizes) and CROQ (for refining questions using prediction sets). Reviewer pywK found CROQ particularly interesting, highlighting its surprising ability to guide LLMs toward the correct answer, and noted its connection to concepts like chain-of-thought reasoning.

**2. Clarity (T1gy, yss3, mTL6, pywK)**

All reviewers commended the clarity and accessibility of the paper. They noted that our methods and motivations are well-explained and easy to follow.

**3. Thorough empirical evaluation and results (T1gy, yss3, mTL6, pywK)**

Reviewers appreciated the thorough empirical evaluation of our methods across multiple datasets and models, with mTL6 and pywK specifically highlighting the accuracy improvements achieved through the CROQ procedure.



### **Response to Queries**


**1. Lack of coherence in the methods (yss3, pywK)**

We understand the reviewers' concerns about the perceived separation between CP-OPT and CROQ. While the methods can function independently, they are complementary and align with our broader goal of robust uncertainty quantification and accuracy improvement in LLMs. We have included the use of CP-OPT and CROQ together as Hypothesis H3. To support this hypothesis, we have added new experiments (Tables 11-13), demonstrating how CP-OPT enhances CROQ by producing smaller, high-coverage prediction sets that improve LLM performance in the second round of querying. These updates clarify the synergy between the methods.

**2. Clarification on connections to Monty Hall (mTL6, pywK)**

The Monty Hall analogy is used to provide an intuitive framework for understanding CROQ’s effectiveness. In CROQ, the conformal set acts as a probabilistic "oracle," eliminating incorrect answers with high probability (e.g., 95%) while ensuring the correct answer remains. This allows the LLM to re-evaluate its predictions with a refined set, sometimes improving accuracy. Unlike the traditional Monty Hall setup, this oracle’s knowledge is derived from conformal scores, which can be generated by the same LLM or external models, offering flexibility. We have clarified this analogy in the paper.

**3. Novelty (yss3)**

Our methods are novel in improving uncertainty quantification and accuracy of LLMs in settings with a finite number of response options (such as MCQs). Prior works rely on heuristic scores or LLM logit scores which could be poorly calibrated or produce larger sets than necessary to achieve the target coverage. In contrast, our work introduces a principled framework CP-OPT for learning optimal scores and CROQ for leveraging refined prediction sets obtained by running conformal prediction on CP-OPT or logit scores to enhance decision-making. We provide a detailed response on this in the response to reviewer yss3.

**4. Results on set size reduction (mTL6)**

We provided additional results (Table 10, Figures 6-13) to aid in understanding the effectiveness of CP-OPT. These visualizations show how CP-OPT redistributes prediction set sizes, reducing the proportion of larger sets while increasing smaller sets. This redistribution is crucial for improving the CROQ and deferral tasks, as smaller set sizes lead to better outcomes. We also discuss factors influencing CP-OPT's performance, such as features and calibration sample sizes.


### **Additional Experiments and Results**


**1. Experiments on more models (T1gy)**

We conducted additional experiments on the gemma-2-9b-it-SimPO model (Meng et al., 2024) using the MMLU dataset with 4, 10, and 15 response options (Tables 10–13 in the Appendix). These results validate our hypotheses:

- **H1:** CP-OPT reduces average set sizes significantly while maintaining comparable coverage (e.g., the proportion of sets of size 15 drops from 41.7% with logits to 25.96% with CP-OPT in Table 11).
- **H2:** CROQ improves accuracy, particularly for smaller sets.
- **H3:** CROQ with CP-OPT scores outperforms CROQ with logits by leveraging smaller, high-coverage prediction sets.

These results confirm the robustness of our methods and address the reviewers' request for evaluation on additional models. Moreover, these results provide further evidence in support of the main hypotheses in the paper.


--------


[ Response continues in the next comment ]

---

> ### Author Response · Authors · 2024-11-23
> **Common Response (Part 2)**
>
> **2. Visualizations for set size distributions**
>
> We added histograms visualizing the set size distributions for CP-OPT and logit scores (Figures 6–13 in the Appendix). These figures show that CP-OPT consistently reduces the proportion of large sets while increasing smaller ones, aligning with its design to minimize uncertainty. This redistribution explains the observed improvements in CROQ and deferral tasks.
>
>
> **3. Impact of set size reduction on accuracy and motivation for CP-OPT in CROQ**
>
> We conducted an additional experiment (Figure 4) to demonstrate how reducing prediction set size improves accuracy with CROQ. Using the Truthful QA dataset with 15 response options, we generated conformal prediction sets based on logits and simulated varying levels of set size reduction by using ground truth to eliminate 0 to 10 incorrect answers while maintaining a constant coverage of 95%. The results reveal a clear trend: as more incorrect answers are removed, the LLM's accuracy after requerying consistently improves. This underscores the need of a score function like CP-OPT, which minimizes set size while preserving coverage, helping CROQ to boost accuracy.
>
>
> We look forward to discussions and addressing any additional questions.

---

### Meta-Review · Area_Chair_JLEy · 2024-12-23

**Metareview:**

This paper presents two methods: CP-OPT, which trains an optimal score function and threshold used for conformal prediction in the LLM settings, and CROQ, which rephrases a multi-choice question by eliminating some choices ruled out by methods like conformal prediction or setting a threshold on logits, etc. The goal of CP-OPT is to perform uncertainty quantification and CROQ is to enable LLMs to make better decisions in tasks like multi-choice question answering.

The paper is interesting, well-written and easy to follow. There are many evaluation experiments to validate the hypotheses the authors had regarding the performance of the two proposed approaches. I think there is enough coherence in the two methods, since uncertainty quantification methods need to be evaluated on downstream tasks that use the predicted uncertainty levels. This coherence is not clear in the current writeup since the authors emphasized uncertainty quantification as the motivation and decision making becomes an add-on. If the authors focus on the decision making motivation and then talk about uncertainty quantification, the methods may appear more coherent.

I don't think Monty Hall reflects the essence of the introduced uncertainty quantification or decision making mechanisms. Some reviewers also expressed similar doubts. The Monty Hall problem represents a paradox that it's always good to change the original choice made by a person if certain new information becomes available. However, in this paper, the original choice made by the LLM plays no role in the rephrased question. The rephrased question seems to depend only on the uncertainty predictions made by an oracle. If this paper were to keep "Monty Hall" as an analogy, at the very least the choice made by the LLM before rephrasing should be taken into account in the next decision, e.g., removing the LLM's original choice in the rephrased question. However, it's quite likely that removing the LLM's original choice will lower the performance. Hence, I believe it's in the best interest of the authors to remove this analogy.

Some points good to clarify if the authors want to improve the paper:
- For an alpha value different from the one used for CP-OPT, whether we have to learn the score function and threshold value from scratch again.
- For CROQ with logits, how the threshold was tuned, and how the threshold can be learned from training data.
- Emphasize that distribution-free does not mean there is no assumption that the train, test and validation data need to follow the same distribution. But instead, it just means we don't need to know which exact distribution the datasets are sampled from.
- How the theoretical guarantees transfer from CP to CROQ.

During the AC-reviewer discussion period, Reviewer mTL6 shared that they believe there are enough ideas, but the the paper was not well put together. "The first part on reducing the number of answers to achieve a desired coverage only showed slight improvement experimentally. The second part on improving performance by a two stage approach showed reasonable improvement experimentally but is only weakly tied to the first part. Furthermore the mechanism for improvement is not clear and I think their Monty Hall analogy is misleading as the mechanism for improvement is probably different from the mechanism in the Monty Hall problem."

Reviewer yss3 and pywK both expressed remaining concerns about the novelty and it'd be great if the authors could clarify more. Reviewer T1gy shared that "recent literature have mostly focused on open-ended generation tasks. Hence the inclusion of conformal prediction towards MCQ decision making is promising (This point is also emphasized in their related work). To emphasize on the experiments, I found the results to be promising across different baselines."

Reviewer pywK shared the major concern about the soundness of QROC: "The authors use conformal prediction to estimate a quality threshold for eliminating low-quality answers. However, the authors conceded that one could simply estimate the quality threshold directly, without using conformal prediction, and achieve (at least) the same performance. The only possible justification for QROC would be if it provides guarantees about the worst-case performance of the downstream predictor (or something similar). However, the authors were unable to develop any meaningful guarantees during the rebuttal." I think the authors might be able to show some guarantees on the final decision making performance by assuming the LLM makes random decisions. But this is not true in practice. If the authors can show a pre-trained or instruction-tuned LLM actually makes the right decision, it'll be a huge contribution to the field.

I think there are more weaknesses than strengths and hence recommend rejection.

**Additional Comments On Reviewer Discussion:**

During the AC-reviewer discussion period, Reviewer mTL6 shared that they believe there are enough ideas, but the the paper was not well put together. "The first part on reducing the number of answers to achieve a desired coverage only showed slight improvement experimentally. The second part on improving performance by a two stage approach showed reasonable improvement experimentally but is only weakly tied to the first part. Furthermore the mechanism for improvement is not clear and I think their Monty Hall analogy is misleading as the mechanism for improvement is probably different from the mechanism in the Monty Hall problem."

Reviewer yss3 and pywK both expressed remaining concerns about the novelty and it'd be great if the authors could clarify more. Reviewer T1gy shared that "recent literature have mostly focused on open-ended generation tasks. Hence the inclusion of conformal prediction towards MCQ decision making is promising (This point is also emphasized in their related work). To emphasize on the experiments, I found the results to be promising across different baselines."

Reviewer pywK shared the major concern about the soundness of QROC: "The authors use conformal prediction to estimate a quality threshold for eliminating low-quality answers. However, the authors conceded that one could simply estimate the quality threshold directly, without using conformal prediction, and achieve (at least) the same performance. The only possible justification for QROC would be if it provides guarantees about the worst-case performance of the downstream predictor (or something similar). However, the authors were unable to develop any meaningful guarantees during the rebuttal."

---

### Decision · Program_Chairs · 2025-01-22

Reject